# The Power of Power Law: Asymmetry Enables Compositional Reasoning

**Zixuan Wang** [1]  **Xingyu Dang** [1]  **Jason D. Lee** [2]  **Kaifeng Lyu** [3]

## Abstract

Natural language data follows a power-law distribution, with most knowledge and skills appearing at very low frequency. While a common intuition suggests that reweighting or curating data towards a uniform distribution may help models better learn these long-tail skills, we find a counterintuitive result: across a wide range of compositional reasoning tasks, such as state tracking and multi-step arithmetic, training under power-law distributions consistently outperforms training under uniform distributions. To understand this advantage, we introduce a minimalist skill-composition task and show that learning under a power-law distribution provably requires significantly less training data. Our theoretical analysis reveals that power law sampling induces a beneficial asymmetry that improves the pathological loss landscape, which enables models to first acquire high-frequency skill compositions with low data complexity, which in turn serves as a stepping stone to efficiently learn rare long-tailed skills. Our results offer an alternative perspective on what constitutes an effective data distribution for training models.

## 1. Introduction

In many domains of machine learning, including vision and language, the model performance often has been observed to follow a power-law scaling with respect to dataset size and model size (Kaplan et al., 2020; Hoffmann et al., 2022; Sorscher et al., 2022; Hestness et al., 2017; Gordon et al., 2021; Henighan et al., 2020). A common hypothesis is that this phenomenon arises from heavy-tailed structure in the underlying data distribution. At the lexical level, natural language exhibits Zipf's law in word frequencies (Zipf, 2016). At a more abstract level, language data may be viewed as

[1]Princeton University [2]University of California, Berkeley [3]Tsinghua University. Correspondence to: Kaifeng Lyu <klyu@mail.tsinghua.edu.cn>.

*Proceedings of the 43rd International Conference on Machine Learning*, Seoul, South Korea. PMLR 306, 2026. Copyright 2026 by the author(s).

consisting of many latent "skills" or "knowledge pieces" whose occurrence frequencies follow a power-law distribution, $p_i \propto i^{-\alpha}$ for some $\alpha > 0$. This perspective has been made more concrete in recent studies that attempt to quantify discrete knowledge and skills in language models (Michaud et al., 2023; Arora & Goyal, 2023).

Under such a distribution, a power-law learning curve may naturally arise when increasingly rare knowledge and skills become covered when the dataset scales. However, this perspective also suggests a potential data inefficiency: rare skills are observed only when the dataset becomes very large, while the most frequent skills may be repeatedly sampled far beyond what is necessary for learning them.

This view motivates investigating whether shifting the data distribution, for example, by reweighting existing data or by deliberately curating new data, can help models acquire long-tail knowledge and skills more efficiently and potentially improve upon standard power-law scaling trends (Sorscher et al., 2022; Medvedev et al., 2026b). In particular, a natural approach to consider is to balance the training data by up-weighting low-frequency skills and knowledge pieces while down-weighting high-frequency ones (Jamal et al., 2020; Zevallos et al., 2023). Given sufficient knowledge of the underlying data distribution and adequate compute budget for data curation, one might therefore expect that an ideal data distribution would be close to a uniform distribution over all skills and knowledge components, assigning almost equal probability mass to each.

However, in this paper, we show that shifting towards a uniform distribution may not always be the best choice. We focus on compositional reasoning tasks, where models are required to combine multiple reasoning skills to solve a problem. We start with a simple example, *multi-step arithmetic*, where models are required to apply basic operations (addition, subtraction, multiplication) to specific operands, and these atomic components serve as the underlying skills. For example, for the problem $2 \times 3 + 1 - 4$, the three operations $[\times 3, +1, -4]$ can be seen as skills. Surprisingly, we find that when each individual skill is sampled from the power law distribution in the training set, the model consistently outperforms its counterpart trained with data under uniform sampling, even though the test accuracy is measured under uniform distribution.

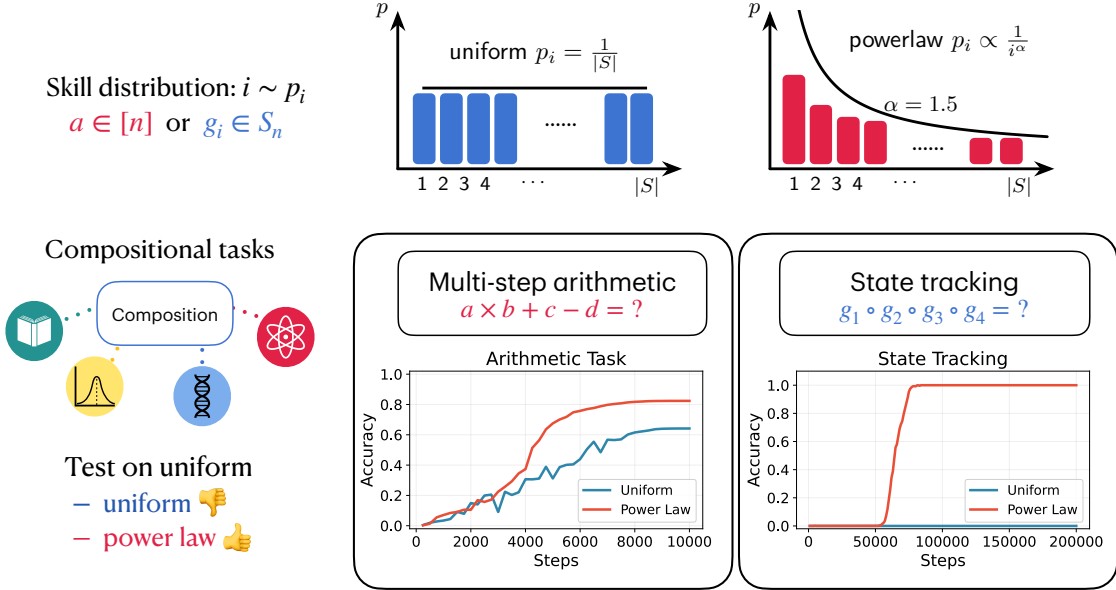

*Figure 1.* Compositional reasoning tasks require composition of skills. We find that only changing the distribution of skills from uniform to power law enables the language models to learn compositional reasoning tasks faster (arithmetic), or even turning an unlearnable task (e.g. state tracking) into a learnable one.

Moreover, the gap is even larger when it comes to *implicit multi-hop tasks* like the *$k$-hop state tracking* task (Merrill & Sabharwal, 2023b; Li et al., 2024b). The models cannot learn without chain-of-thought (CoT) or curriculum learning under uniform distribution, but training with a power law distribution of skills enables the same model to directly learn the task efficiently.

Towards understanding the counterintuitive advantage of power-law distribution, we propose a minimalist task to model the skill composition for theoretical insights. Under this setting, we are able to investigate why the uniform data distribution induces hardness for composition tasks, and how the power-law distribution helps to re-enable the efficient training. Corresponding to the theoretical setting, we also conduct various experiments to mechanistically verify the predictions from the minimalist model and further confirm the advantage of power-law distribution.

**Our contributions** are summarized as follows:

- We observe the counter-intuitive phenomenon on *compositional reasoning tasks* that training on data following an asymmetric power-law consistently outperforms training on uniformly sampled data. Sampling data following power-law distribution can even enable models to learn some complex *implicit multi-hop tasks* efficiently without any intermediate supervision (e.g. curriculum or chain-of-thought), while models cannot learn under a uniform distribution.

- We propose a minimalist theoretical task to model

the general compositional reasoning tasks, called *k-multiplicative composition*. We rigorously prove that if the model is trained under a uniform distribution, learning the task requires at least $d^{\Omega(k)}$ samples or runtime. However, training with gradient descent (GD) under a power-law distribution only requires $d^{O(1)}$ samples and runtime, thus establishing a theoretical separation between uniform and power-law distribution.

- In line with our theoretical analysis, we show that a similar stage-wise learning mechanism is also confirmed in the *state tracking* task. The power-law distribution first significantly improves the pathological landscape near initialization and strengthen the initial learning signals of composition. The high-frequency skills are first learned and then in turn benefit the learning of scarce long-tail skills, with the intuitive long-tail drawback of power-law appearing in the final stage.

- We finally validate our theoretical predictions on more complex synthetic natural language reasoning tasks, including multi-hop question answering (Yao et al., 2025a) and grade-school math problems (Ye et al., 2024; Zhou et al., 2025), empirically showing the advantage of power-law distribution in the training of compositional reasoning tasks.

## 2. Motivating Experiments

We begin by presenting the empirical phenomenon that motivates our theoretical investigation. We focus on *com-*

*positional reasoning tasks*, where a model must compute the result directly by combining several steps of operations. These tasks represent the abundant implicit reasoning within language models, which require the models to understand indivisible or hidden composition of skills or knowledge without explicit thinking process (Michaud et al., 2023; Arora & Goyal, 2023). As one of the most intuitive compositional reasoning tasks, we choose *multi-step arithmetic* as a starting testbed for our observation.

**Multi-step arithmetic.** We first observed the effectiveness of power-law distribution in *multi-step arithmetic* task in Deng et al. (2024); Wang & Lu (2023), where the model must directly output the integer result of an expression containing $k = 4$ sequential operations (including $+, -, \times$) without chain-of-thought. Here the model needs to learn how each operation transforms the current intermediate result, so *each operand combined with the operation (e.g. $+3, -2$) is seen as an atomic skill to sample* in the training distribution. We train a 0.6B-parameter model with Qwen3-style (Yang et al., 2025) architecture from scratch, ensuring that the model starts with no prior knowledge of numerical identities. We surprisingly find that simply switching the sampling strategy from uniform distribution to a power-law distribution with the exponent $\alpha = 1.0$ results in a significant performance improvement, as shown in Figure 1. Note that the **number indices are randomly shuffled**, so the power law governs only the occurrence frequencies without any inherent ordering of the operands.

**State tracking.** To test the generality of the phenomenon, we also experiment with some algorithmic task — the *state tracking task* proposed in Merrill & Sabharwal (2023b) based on the symmetry group $S_5$. State tracking modeling has been long established in both the theoretical and empirical literature in language modeling and sequential reasoning. In this task, the model is required to compose the sequence of input group elements $g_1, g_2, ..., g_k \in S_5$. The target is to output $g_1 \circ g_2 \circ \cdots \circ g_k$ without chain-of-thought, where $\circ$ is the group operation. However, even though there exists an $O(\log k)$-layer transformer that can compose $k$ group elements, the task has been proved to be challenging to learn under *uniform* training distribution without intermediate supervision for all architectures both theoretically (Wang et al., 2025) and empirically (Li et al., 2024b).

Similar to the arithmetic tasks, our experiments in Figure 1 show that a transformer can actually learn the task almost perfectly under the **power-law** distribution ($\alpha = 1.5$) without any curriculum (Wang et al., 2025) or intermediate thinking trace (Li et al., 2024b), while the uniform training distribution cannot. Note that the rank of different group elements $g_i \in S_5$ is also given **randomly** across experiments, so the learning speed-up is not due to the learning order from easier functions to harder functions.

The observations from these two experiments provide surprising evidence that instead of hurting performance due to the long-tail effect, an asymmetric power-law distribution is actually good for the learning of compositional tasks.

## 3. A Minimalist Example of Composition

Towards understanding why only a switch of training distribution helps in implicit compositional reasoning tasks, we aim to find the simplest modeling for skill composition and reveal why uniform distribution may face challenges in training. In this section, we introduce our theoretical setting, together with a lower bound showing that gradient-based training will provably require large amount of data or compute under uniform distribution.

**Notations.** For any $n \in \mathbb{N}$, $[n] = \{1, 2, ..., n\}$. Bold lowercase letters represent vectors (e.g. $e$). Normal lowercase letters are scalars. Bold uppercase letters represent matrices (e.g. $X$). $\|u\|_2$ denotes the $\ell_2$-norm of a vector $u$. For any index $i \in [d]$, let $e_i \in \mathbb{R}^d$ denote the $d$-dimensional one-hot vector with a 1 in the $i$th coordinate. We use $\tilde{O}, \tilde{\Omega}$ to hide $\mathrm{polylog}(d)$ factors, and we use $f \lesssim g$ (or $f = O(g)$, $g = \Omega(f)$) when $f \leq Cg$ for an absolute constant $C > 0$.

### 3.1. $k$-multiplicative Composition

Analyzing transformers trained on real-world compositional reasoning tasks could be challenging due to technical difficulties and many potential entangled factors. Therefore, we propose a minimalist task as the abstraction of multi-hop skill composition to gain theoretical insights for a better understanding of this phenomenon. Similar to the *state tracking* composition tasks in Merrill & Sabharwal (2023b), we consider a sequence to sequence task that requires to compose a sequence of input functions as skills. We assume that $d$ fixed skills $s_1, ..., s_d$ are used in the task, and we fix the number $k$ as the *hop number* for each input sequence.

**Setting.** We consider the following $k$-**multiplicative composition** task as the composition of a sequence of $k$ 'scalar skills' independently sampled from those fixed $d$ available skills. Each skill $s_i$ represents a hidden scalar $w_i^* \in \{-1, +1\}$, and the composition operation of skills is multiplication. Formally, the length-$k$ input sequence $X = (x_1, ..., x_k)$ is a sequence of skill vectors sampled from $\{e_i : i \in [d]\}$, which are one-hot vectors representing different $d$ skills. The ground-truth label $y$ is given by the composition of the hidden scalars behind the input skills $s_i$: $y := \prod_{i=1}^{k}(x_i^\top w^*)$, where $w^* := (w_1^*, ..., w_d^*) \in \mathbb{R}^d$ is the hidden vector, which is the concatenation of the hidden scalars $w_i^*$ of each skill $i$.

The goal of the task is to uncover the hidden scalar $w_i^*$, given the training samples $(X, y)$ sampled from a certain distribution. Specifically, we define the target function class

$$\mathcal{F} = \{f_{\boldsymbol{w}}(\,\cdot\,) : \boldsymbol{w} \in \{\pm 1\}^d\}, f_{\boldsymbol{w}}(\boldsymbol{X}) = \prod_{i=1}^{k}(\boldsymbol{x}_i^\top \boldsymbol{w}).$$

The ground-truth label $y = f_{\boldsymbol{w}^*}(\boldsymbol{X})$ is given by the ground-truth hidden vector $\boldsymbol{w}^*$. Given any learner model $f_\theta(\boldsymbol{X})$, the objective we are required to optimize is the MSE loss $\hat{\mathcal{L}}(\boldsymbol{w}) = \frac{1}{n}\sum_{i=1}^{n} \ell(\boldsymbol{w}; \boldsymbol{X}^{(i)})$, where the per sample loss is

$$\ell(\boldsymbol{w}; \boldsymbol{X}^{(i)}) = \frac{1}{2}\big(f_\theta(\boldsymbol{X}^{(i)}) - y\big)^2$$

for each sample $\boldsymbol{X}^{(i)} = (\boldsymbol{x}_1^{(i)}, ..., \boldsymbol{x}_k^{(i)})$. The training distribution of the sequence is defined as follows: each input skill $\boldsymbol{x}_j^{(i)} \overset{i.i.d.}{\sim} \mathcal{D}_{\text{train}}$ is independently sampled from a certain fixed distribution $\mathcal{D}_{\text{train}}$. In this work, $\mathcal{D}_{\text{train}}$ can be uniform over the skills $\text{Unif}(\{\boldsymbol{e}_1, \boldsymbol{e}_2, ..., \boldsymbol{e}_d\})$ or a certain power law $\Pr[\boldsymbol{x}_i = j] = p_j$ with $p_j \propto j^{-a}$ where $a > 0, j \in [d]$.

**Remark.** The task can be seen as a generalization of the parity task. Instead of directly computing the product of input scalars $\{-1, +1\}$ like parity task itself, the model has to learn the hidden knowledge $w_i^*$ behind the input skill. The model itself can also be seen as a recurrent neural network (RNN) with a scalar hidden state $h_i \in \mathbb{R}$ and parameter $\boldsymbol{w} \in \mathbb{R}^d$, with the input sequence $(\boldsymbol{x}_1, \boldsymbol{x}_2, ..., \boldsymbol{x}_k)$ and the update rule $h_{i+1} = g_{\boldsymbol{w}}(\boldsymbol{x}_i, h_i) = (\boldsymbol{x}_i^\top \boldsymbol{w})h_i, h_1 = 1$. This data model is also similar to the sequence single index model proposed in Arnaboldi et al. (2025) and the PolyNet in Barak et al. (2022).

### 3.2. Uniform distribution fails: lower bound

Can gradient-based algorithm learn the task efficiently? Similar to many algorithmic tasks like $k$-sparse parity or $k$-fold composition (Szörényi, 2009; Wang et al., 2025), we show that any learner must either use a large amount of training data or runtime when the training distribution is uniform.

Formally, we prove a correlational statistical query (CSQ) lower bound (Szörényi, 2009) for learning $\mathcal{F}$. Online stochastic gradient descent (SGD) on the square loss is included in the CSQ learner class. The discussion and the proof is in Section A.2. Our lower bound against the CSQ learner is given as follows:

**Theorem 1** (Informal). *Let the input distribution be uniform. There exists a constant $\epsilon = \Omega(1)$, s.t. any model trained with gradient descent using $q$ gradient queries requires a tolerance $\tau^2 \le \left(\frac{\log(dq)}{d}\right)^{k/2}$ to achieve loss $\mathcal{L}(\boldsymbol{w}) \le \epsilon$, which requires $n \gtrsim d^{k/2}$ samples when $q \lesssim d^{k/2}$.*

The theorem implies a learner must use either enormous compute or sample size of $\tilde{\Omega}(d^{k/2})$, which is unacceptable when the number of skills $d$ is large and multiple hops of

composition are required. Therefore, training under uniform distribution suffers from the computational gap and fails on such compositional task.

That said, the existing lower bound restricted the training **distribution**: the proof only holds when the distribution is uniform. The key insight behind such computational hardness lies in the symmetry of the function class, causing it hard to distinguish from one function from another within the function class. Then the natural question arises: *does the asymmetry in power law help to break this lower bound?*

## 4. A Theory for Why Power Law Helps

In this section, we show that power-law distribution indeed helps in the training on the minimalist setting. We prove that with online SGD, the model learns the ground-truth skill vector $\boldsymbol{w}^*$ with much smaller sample size. Based on the theory insights, we summarize three stages in the learning process of the compositional tasks, and try to mechanistically verify the theoretical insights with experiments on the state tracking task.

### 4.1. Power-law distribution enables composition

We first show a positive optimization result on the power-law distribution. In contrast to uniform distribution, it is possible that online SGD can learn the target function under the power-law distribution, with prior knowledge of the composition structure. Specifically, we simply consider the learners that take a similar form:

$$f_{\boldsymbol{w}}(X) = \prod_{i=1}^{k}(\boldsymbol{x}_i^\top \boldsymbol{w}), \text{ where } \boldsymbol{w} \in \mathbb{R}^d,$$

and we optimize on the parameter $\boldsymbol{w}$. The goal is to recover the ground truth $\boldsymbol{w}^* \in \{-1, 1\}^d$.

The following theorem shows that if we pick the power-law distribution as the training distribution $\mathcal{D}_{\text{train}}$, online SGD can learn the ground truth with much less samples $\tilde{O}(d^{2\alpha})$ compared to the lower bound of using uniform distribution.

**Theorem 2** (Gradient descent learns under power law). *Let the input distribution be Zipf law with $p_j \propto j^{-\alpha}$ with $\alpha > 1$. Suppose that the target error is $\varepsilon > 0$, and $\boldsymbol{w}(0) \sim \mathcal{N}(0_d, r^2\boldsymbol{I}_d), r = \Theta(1), k = \Theta(1)$ and is even, $\eta \le O\left(\frac{1}{k^2\|\boldsymbol{p}\|_2}\right)$ and $B \ge \tilde{O}\left(\frac{d^\alpha \log \frac{1}{\delta}}{p_{\min}\varepsilon}\right)$. Then with probability $1 - \delta$ the model can learn the task by minibatch gradient descent with $\tilde{O}(\frac{d^{2\alpha}}{\eta\varepsilon})$ samples in $t \le \tilde{O}(\frac{d^\alpha}{\eta}\log\frac{1}{\varepsilon})$ time, with error $\min\{\|\boldsymbol{w} + \boldsymbol{w}^*\|_\infty, \|\boldsymbol{w} - \boldsymbol{w}^*\|_\infty\} \le \varepsilon$ and the population loss $\mathbb{E}[\mathcal{L}(t)] \le O\big(d^{-\alpha}\varepsilon^2\big)$.*

When the composition number is large compared to the power-law distribution exponent (e.g., $k \ge 4\alpha$), we have a sample complexity or runtime improvement using power-

law distribution compared to the uniform distribution. The proof is deferred to Section A.3.

**Key Proof Idea.** The proof idea is based on the gradient descent dynamics on the population loss, which is the expectation of the training loss. After some simplification, the first step update is close to the negative expected gradient $-\nabla_{\boldsymbol{w}}\mathcal{L}_{\mathcal{D}}$ at initialization:

$$w_j(1) - w_j(0) \approx -\eta\nabla_{w_j}\mathcal{L}(\boldsymbol{w}(0))$$
$$= \eta k p_j\big(A(0)^{k-1}w_j^* - B(0)^{k-1}w_j(0)\big),$$

where $A(t) = \sum_{i=1}^d p_i w_i(t)w_i^*$, $B(t) = \sum_{i=1}^d p_i w_i(t)^2$ are the weighted inner product and weighted norm with respect to the power law probability distribution. While with the power-law distribution, the probability $p_i$ of 'head' skills that has a constant rank $i = O(1)$ is also constant. With an initialization scale of small constant $r$, $|A(0)| \approx \Theta(r) \gg |B(0)| \approx \Theta(r^2)$. Therefore, if we only keep the larger term $p_j A(0)^{k-1}w_j^*$, the initial gradient is approximately

$$\nabla_{\boldsymbol{w}}\mathcal{L}(\boldsymbol{w}) \approx -k\,\mathrm{diag}(\boldsymbol{p})A(0)^{k-1}\boldsymbol{w}^*,$$

which is a constant large initial gradient for the head skills. That actually indicates both good initial landscape and even global loss landscape. With this initialization, we can prove a Polyak–Łojasiewicz condition (Karimi et al., 2016) s.t.

$$\|\nabla\mathcal{L}(\boldsymbol{w}(t))\|_2^2 \gtrsim p_{\min}A(0)^{2k-2}\mathcal{L}(\boldsymbol{w}(t)).$$

This guarantees the convergence of population GD dynamics in $\tilde{O}(\frac{1}{\eta p_{\min}}\log\frac{1}{\varepsilon})$ time for $\varepsilon$ error. With the population dynamics, we can apply finite sample concentration analysis and prove the desired sample complexity requirement.

**Remark.** Note that the proof technique also apply for uniform distribution. However, under the uniform distribution, the probability $p_i = \frac{1}{d}$ for each skill leads to a very small initial $A(0) \approx O(\frac{1}{\sqrt{d}})$, which makes the initial gradient norm $\frac{1}{d^{\Omega(k)}}$. Thus, it takes $d^{\Omega(k)}$ time to escape from the initialization and leading to $d^{\Omega(k)}$ sample complexity.

### 4.2. Mechanistic understanding of the learning stages under power law

Taking one step further at the theoretical training dynamics, we can observe some stage-wise phenomenon in the training process. First, the power-law distribution improves the initial landscape and leads to faster escape from the flat initial region. After escaping the flat region, the head skills (e.g. $w_i^*$ with a constant skill index $i = O(1)$) are learned first at a faster rate, which in turn accelerates the learning process of the tail skills (e.g. $w_j^*$ with a large skill index $j = \Omega(d)$). In the final stage, all the skill hidden scalars are learned, but got slowed down by the low long-tail probability of

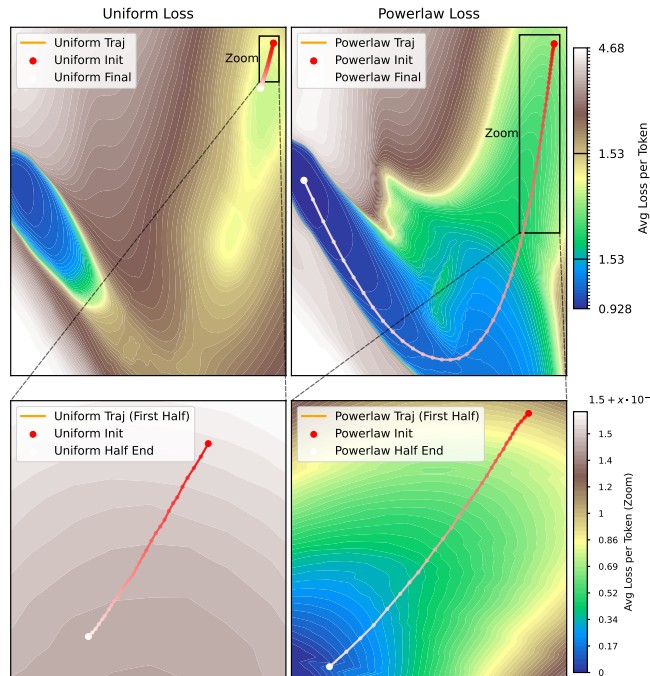

*Figure 2.* **Initial loss landscape comparison.** We plot the loss landscape under both uniform and power-law distribution, where the two directions are determined by PCA of both trained checkpoints along the trajectory. The training loss landscape based on power-law distribution has an apparently steeper slope as the descent direction, while training on uniform distribution fails to escape from the initial flat region.

sampling the tail skills. We further verify the stage-wise dynamics characterization with experiments and visualization on the $S_5$ state tracking task with the hop number $k = 4$ to confirm the generality of the theoretical insights.[1]

**Stage I: Power law enables escaping from flat region.** As mentioned in Section 4.1, the initial population gradient of the task is much larger when the training distribution follows power law compared to a uniform distribution. The larger gradient signal indicates a improved loss landscape via changing the training distribution, where a clearer descent direction to the lower loss region should exists. In contrast, the initial region of the loss of uniform distribution should be far flatter with a much smaller slope.

To verify the theoretical prediction, we take the training trajectories of the $S_5$ state tracking composition task, both power law and uniform distribution. We analyze the top-2 principal components of the difference between consecutive logged checkpoints $\theta_t - \theta_{t-\Delta t}$, and plot the loss landscape together with the training trajectory in Figure 2. Since the region near initialization is very flat, we zoom in the the

---

[1]In this subsection, the order of the permutations used in the power law follows the lexicographical order. This is different from other experiments in this paper, where the order of skills are randomly ranked. We will discuss the order in Section 4.3.

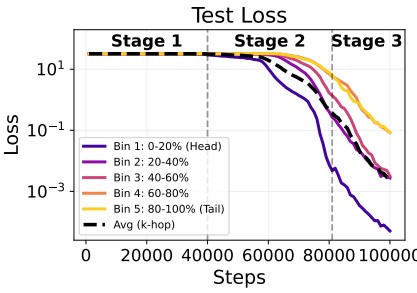 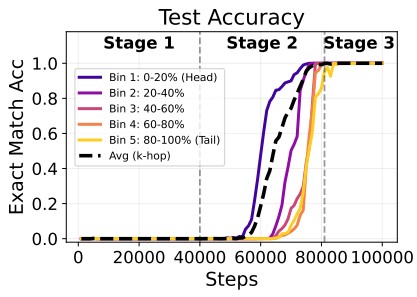 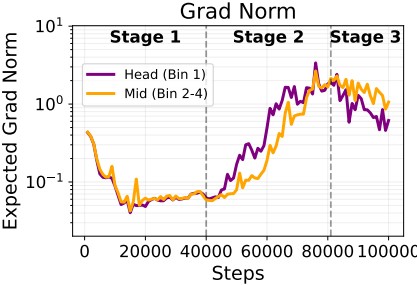

*Figure 3.* **Training dynamics under power law loss on** $S_5$**. Left:** Test loss in total and on subset with samples composed from different group of permutations (ordered by rank). **Mid.** Test accuracy of each group. **Right.** The gradient norm on samples that requires tail skills. When the head skills are learned after stage 1, the gradient norm increases and speeds up the learning of tail skills.

initial region and draw denser contours near initialization for better visualization. The experiments verified our prediction: the power-law distribution induced a much better initial loss landscape, while training loss with uniform distribution is much flatter and harder to optimize by gradient methods.

**Stage II: Head skills help the tail.** Though the initial gradient signal is large enough to escape the flat region, the hidden scalars behind the skills are not learned simultaneously. Actually, the head skills are learned first, which further accelerates the training of the tail. Now we consider the initialization scale as a small constant $r \ll 1$. Recall the expected gradient update (negative population gradient):

$$w_j(t+1) - w_j(t) = \eta k p_j \left( A(t)^{k-1} w_j^* - B(t)^{k-1} w_j(t) \right)$$

According to the gradient formula, the head skills with $p_i = \Theta(1)$ will grow to a large constant at first from the initialization scale $r$. That will increase the value of the weighted similarity $A(t)$ from $O(r)$ to a large constant $O(1)$, which significantly increases the signal term $k p_j A(t)^{k-1} w_j^*$ in the gradient. So after the initial escape from the flat region, the power law induces an implicit curriculum enables fast learning of high-frequency skills, which helps the training on the scarce long-tail skills in return.

Experiments on $S_5$ also confirm the acceleration effect after the first stage, as shown in Figure 3. We separate the group elements (i.e. permutations) in the order of rank into five bins, with rank 0%–20%, 20%–40%, etc. After the test accuracy of samples where all $k = 4$ elements are in the first bin exceeds a threshold (0.1%), we consider the training enters stage 2. To check if the learned skills help to accelerate the training of the tail skills, we measure the gradient norm of samples with only one input permutation in the tail bin, while others are sampled from other bins of input permutations. We compare the following two possible sample sets to compute the expected gradient: (1) all three other permutations are in bin 1, where all the head permutations lie in; (2) the other three permutations are sampled from bin 2 to 4. As shown in Figure 3 (Right), with head input permutations of Bin 1 in the input sequence, the expected gradient norm

of such samples is much larger than the gradient norm of the second group in stage 2. That confirms that once the head skills are learned, they can indeed make the learning of the tail skills much faster in the compositional tasks.

**Stage III: Long-tail convergence.** When all the skills are learned from scratch and get non-trivial accuracy, the training eventually enters the convergence phase. In this stage, our original intuition finally comes in: the long-tail effect of the rare skills causes slow final convergence of the training loss, since they appear at a much lower frequency. In the theory setting, skills with rank $\gtrsim \Omega(d)$ all have small probability $O(\frac{1}{d^\alpha})$ of being sampled, which indeed slows down the final convergence. Similar phenomenon is observed in the state tracking task. The test loss on the composition tasks on the tail permutation (Bin 4, 5) grows much slower than the head (Bin 1). Despite the final training inefficiency, the first two stages already ensure the successful learning of compositional reasoning tasks.

### 4.3. Ablation Studies

**Effect of the exponent $\alpha$.** We first ablate on the exponent $\alpha$ to see its effect on the training speed up. According to the theory, we need $\alpha > 1$ as a sufficient condition for efficient convergence on $k$-multiplicative composition task. Here, we explore the generality and necessity of different $\alpha > 0$ to test the general effectiveness of the power-law distribution. The experiment details and discussions are in Section C.1. The takeaways are (1) *large enough $\alpha$ is necessary* for efficient learning for hard composition tasks (2) though **larger** $\alpha$ leads to **faster training in the head**, the tail skills' learning will be slowed down due to smaller sampling probability. The result echoes our theory and indicates that there is a trade-off on the exponent $\alpha$.

**The granularity of asymmetry.** Our results show that power-law distribution is a sufficient condition for successful training on compositional reasoning tasks. Although the analysis does not rule out other asymmetric distributions that enable LLMs to acquire composition capabilities, we conjec-

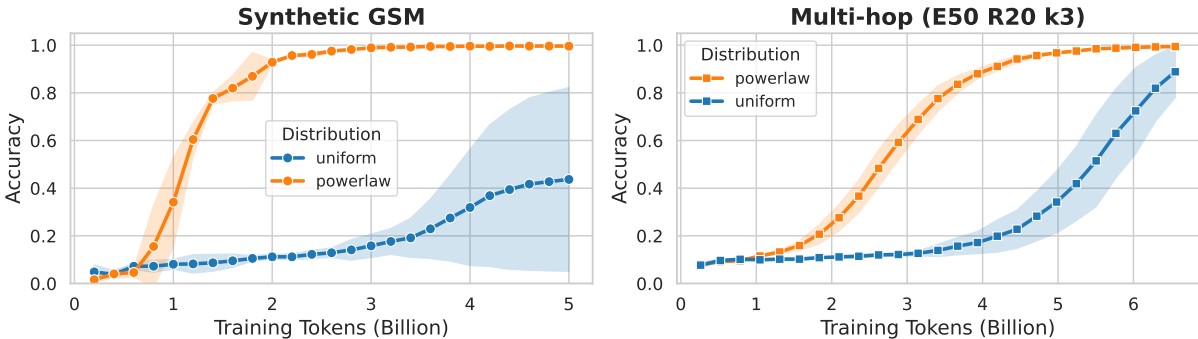

*Figure 4.* **Left.** Test accuracy on synthetic iGSM data. The operations are restricted within 2-8. All arithmetic calculation are done with modulo $p = 211$. **Right.** A multi-hop task with $|E| = 50$ individuals, with each person has $|R| = 20$ relations, and with hop $k = 3$.

ture that *better 'granularity of asymmetry' may lead to better training loss landscape, which further accelerates training.* Power law is an example of fine-grained asymmetry. We also tried several different, more coarse-grained power-law distributions in Section C.4. To be specific, we divide the $|S_5| = 120$ permutations into $m = \{5, 10, 20, 40, 60\}$ bins in lexicographical order, assign the sum probability for different bins with the power law, but keep the individual skill probability in each bin uniform. The larger $m$ is, the distribution is more fine-grained and closer to original power law. The initial experiments show that more fine-grained the distribution is, the faster the model learns. We leave a precise study on the effect of granularity to future work.

**The order of skills.** Note that the skills are similar but not entirely equivalent: there are some 'easy' or more 'fundamental' skills, such as the identity permutation in $S_5$. The order of different skills in the power law sampling process may lead to different results. For example, putting easier skills at the beginning will significantly increase the skill frequency, which may induce an implicit easy-to-hard curriculum and further improve the performance. In Section C.2, we show that (1) *the order matters*: default lexicographical order of the numbers or permutations learn much faster with the same exponent $\alpha$; (2) *power law significantly helps optimization* under a **random** order of permutations, or even a reversed lexicographical order. In summary, the asymmetric power law still accounts for the improvement that makes the state tracking composition task learnable, while the advantage can be strengthened by a designed/structured order of skills. The experiments across multiple different order of permutations rules out the possibility that the benefits come from a special ordering and verifies the effectiveness of the data asymmetry itself.

**Compatibility with explicit curriculum.** We further show that the power-law distribution is naturally compatible with certain curriculum learning and may lead to a potentially better training strategy. In Section C.3, we experiment

with an explicit curriculum based on the number of hops $k = 1, 2, 3, 4$ following Wang et al. (2025), which creates an easy-to-hard supervision over the task complexity. Besides the uniform distribution baseline, we also try a power law distribution experiment to test the conjecture. The experiments show that even with a curriculum, training with uniform distribution still exhibits noticable loss plateaus during training. By contrast, the power law training distribution further enhances the training and substantially reduces the plateaus, leading to faster training by improving the loss landscape along the training trajectory.

## 5. Experiments

To further test our understanding, we conduct experiments on several natural language reasoning tasks to test the advantage of power-law distribution. We considered two natural language synthetic tasks, Multi-hop Question Answering (Yao et al., 2025a) and Synthetic Grade School Math problems (Ye et al., 2024; Zhou et al., 2025). In all experiments, only the training distribution is altered from uniform distribution to a power-law distribution. The test sets are all sampled from the uniform distribution of skills. Without specification, we pick the exponent $\alpha = 1$ for the power law. The order of different skills is chosen randomly, which is used in the power law sampling process.

**Multi-hop QA.** We first consider a natural language reasoning task proposed in Yao et al. (2025a) as a testbed of implicit reasoning capabilities of language models (Yao et al., 2025a; Wang et al., 2024; Ye et al., 2026). The task is based on synthetic facts on relations $\mathcal{R}$ (e.g. teacher, instructor, etc.) between $|\mathcal{E}|$ different individuals (Alice, Bob, etc.). As a concrete example, the facts are like *The instructor of Alice is Bob* and *The teacher of Bob is Carol*. The target is to answer multi-hop queries like *'Who is the teacher of the instructor of Alice?'*, where each relation in the question is considered as a single *hop*.

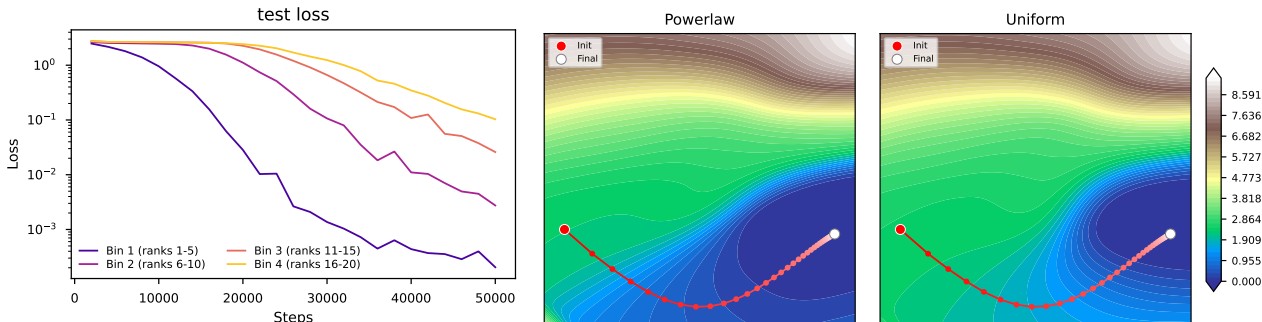

*Figure 5.* **Left.** Test loss of different samples on subset with samples composed from different group of permutations (ordered by rank). **Right.** Loss landscape comparison for multi-hop QA tasks. Power law loss landscape still has a steeper slope, which indicates the generality of the benefit of the power law distribution for training.

We can interpret the relations as a dependency graph: each individual as a node, and each relation as a colored edge. During training, we first fix an underlying dependency graph. The training data is a mixture of the profile fact data (1-hop) and the query questions ($k$-hop). We train a small GPT-like model with online sampled data. The skills we considered are the *relations* $r \in \mathcal{R}$. With a randomly assigned order, we construct QA pairs composing $k$ skills sampled under the power-law distribution.

Yao et al. (2025a) find that transformer-based language models require training data that grows exponentially in $k$ to learn implicit $k$-hop reasoning without curriculum, where each *hop* of the query is sampled uniformly. Our experiments in Figure 4 (right) show that with the power-law distribution, the models learn much faster across different numbers of entities $|E|$ and hop number $k$ with fewer samples, exhibiting the empirical advantage of the power-law distribution in implicit multi-task reasoning tasks. Moreover, we plot the test loss on subset with samples composed from different groups of permutations (ordered by the probability assigned by the power law) and the loss landscape with results similar to the experiments in the $S_5$ experiments, which further generalizes our mechanistic findings in the previous synthetic setting. Additional experiments and details are deferred to Section D.

**Synthetic Grade School Math Problems.** We finally test our conjecture on synthetic grade-school level math problems in natural language. Since grade school math problems often involve multiple variables with some simple dependencies, each problem can be seen as a composition of basic arithmetic operations on the dependency graph. Following Ye et al. (2024) and Zhou et al. (2025), we consider a layered structure of categories and items for problem generation, and map the graph to natural language through synthetic templates. For simplicity, we restrict the dependency graph to 8 operations. We consider each number as a skill for power-law distribution sampling. We consider both arithmetic computation with and without modular operation with

a prime number $p$ in each problem. Section E has more details on the data generation process.

As our experiments show in Figure 4 (left), the model learns much faster with the underlying power-law distribution compared with uniform sampling on the numbers. The advantage of the power-law training generally holds true whenever the arithmetic has modulo $p$ or not. We also considered a synthetic template introducing some calculation combining two consecutive steps to mimic real world thinking trace. As shown in Section E, the model trained under the power-law distribution always outperforms their counterparts trained with uniform distribution with or without such multi-hop structure, showing the robustness of the improvement brought by the power-law distribution.

## 6. Related Works

**Compositional reasoning in LLMs.** Numerous tasks in natural language involves complex composition of atomic skills (Arora & Goyal, 2023; Zhao et al., 2024; Chen et al., 2023; Liu et al., 2025; Ortiz-Jimenez et al., 2023) and involves multi-hop reasoning (Zhong et al., 2023; Zhang et al., 2024; Ju et al., 2024) where most of the composition structure is implicit. Recent works have shown that language models struggle in implicit reasoning without the intermediate thinking process or chain-of-thought (CoT) (Yang et al., 2024; Allen-Zhu & Li, 2023; Press et al., 2023; Kassner et al., 2020; Yao et al., 2025a). Allen-Zhu & Li (2023) showed that language models struggle to solve many simplest multi-hop knowledge manipulation tasks. Yao et al. (2025a) also confirmed the inherent hardness in multi-hop QA tasks, implying exponentially many QA-pair data are required without CoT.

**Hardness of learning composition.** Several lines of literature tried to understand the difficulties in compositional reasoning capabilities of language models and the possible solutions to this issue (Wen et al., 2024; Kim & Suzuki, 2024; Huang et al., 2025b;a; Yao et al., 2025a; Wang et al.,

2024; Ye et al., 2026; Wang et al., 2025). From the theoretical side, many symbolic synthetic tasks (Li et al., 2024b; Merrill & Sabharwal, 2023a; Liu et al., 2023; Merrill & Sabharwal, 2023b; Sanford et al., 2023; Wang et al., 2025; Peng et al., 2024; Chen et al., 2024; Ren et al., 2024) are proposed to model compositional tasks, aiming to understanding difficulties both from the expressiveness (Peng et al., 2024; Chen et al., 2024) and the learning perspective (Wen et al., 2024; Kim & Suzuki, 2024; Wang et al., 2024). Specifically, lower bounds on compositional reasoning tasks under *uniform distribution*, such as parity (Wen et al., 2024; Kim & Suzuki, 2024) and $k$-hop state tracking task in Li et al. (2024b); Wang et al. (2025); Huang et al. (2025b), show that either exponentially many data or training steps are required to learn the task without any intermediate supervision or CoT. Our work reveals that the underlying distributional assumption (i.e. uniform or isotropic distribution) is essential in those lower bounds. We show that breaking the symmetry of the training distribution (e.g. using a power-law distribution) actually enables the model to learn the hard functions without intermediate supervision.

Another series of efforts tried to approach the problem via controlled experiments and mechanistic understanding (Yao et al., 2025a; Wang et al., 2024; Ye et al., 2026; Biran et al., 2024; Li et al., 2024a; Yao et al., 2025b; Kassner et al., 2020). Wang et al. (2024) and Ye et al. (2026) showed that transformer-based language models can only compositionally generalize to multi-hop queries, but cannot naturally compose atomic facts nor generalize to out-of-distribution data that did not appear in the multi-hop training data. Yao et al. (2025b) have found that the atomic knowledge is stored in different layers when the knowledge is used in different hops of composition, leading to the failure of generalization in composition. Architectural explorations on parameter sharing (Wang et al., 2024; Zhu et al., 2025) can mitigate some of the issues but they cannot completely solve the training difficulties.

**Asymmetric data breaks hardness.** Before LLMs, many papers have theoretically investigated the learning dynamics of shallow neural networks on synthetic tasks like single/multi-index models, parity, and other boolean functions, mostly under the assumption of isotropic and uniform distribution (Damian et al., 2022; Barak et al., 2022). Under such data distribution assumption, intermediate supervision like curriculum (Abbe et al., 2021; 2022; 2023) are necessary to learn the target function efficiently.

Some recent works have moved beyond the standard isotropic/uniform distribution and showed potential improvement for more efficient learning (Daniely & Malach, 2020; Cornacchia & Mossel, 2023; Mousavi-Hosseini et al., 2023; Cornacchia et al., 2025). For the learning parity task, Daniely & Malach (2020); Cornacchia & Mossel (2023)

exhibited that a biased distribution can enable more efficient learning than uniform inputs. For learning single index models, Mousavi-Hosseini et al. (2023) demonstrated that a Gaussian training distribution with a spike-covariance structure can lead to improved sample complexity guarantee independent of the information-exponent[2]. Cornacchia et al. (2025) further showed that by introducing a symmetry-breaking random perturbation to the data distribution, Gaussian single-index model becomes efficiently learnable. Our results can be seen as a generalization to the compositional tasks with a natural power law data distribution. The results potentially provide guidance for better training recipes for natural language reasoning tasks. Medvedev et al. (2025; 2026a) further found that distribution shift can be beneficial to test performance due to mismatched training proportions.

**Skill composition in LLMs.** Recent works have attempted to understand the skill composition capabilities of LLMs (Didolkar et al., 2024; Arora & Goyal, 2023; Yu et al., 2023; Chen et al., 2023; Zhao et al., 2024; Michaud et al., 2023). Michaud et al. (2023) hypothesize the power-law distribution of the skills or quanta to explain the power law loss scaling curve. Arora & Goyal (2023) established theoretical foundations for the emergence of skills composition. Yu et al. (2023) and Zhao et al. (2024) benchmarked the skill-composition ability of the LLMs and investigated methods to elicit the capability by supervised finetuning. Didolkar et al. (2024) further showed the evidence that LLMs have metacognitive knowledge for skills of LLMs.

## 7. Conclusion and Future Work

In this paper, we provide an alternative perspective on the criteria of selecting effective data distribution on reasoning tasks when training language models. On many synthetic compositional reasoning tasks like state tracking, arithmetic, multi-hop QA, and grade-school math problems, we empirically show that power law significantly accelerates training compared to using a uniform distribution. Our results take the first steps towards understanding why the asymmetry of power law enables learning of complex compositional tasks via theoretical and empirical mechanistic approach.

There are some limitations of this work and future directions. Most of the experiment settings in our paper are based on the pretraining setting on algorithmic or synthetic natural language tasks. We look forward to future explorations on improving real-world skill composition capabilities as in Yu et al. (2023) by reweighting the data distributions to power law, or composing new skills (e.g. agentic skills/tool calls) missing in the natural language data by synthesizing data with certain distributions.

---

[2]A hardness measure of target functions (Dudeja & Hsu, 2018), which is similar to hop number.

## Acknowledgements

We thank Nathan Srebro and Zhiyuan Li at the Toyota Technological Institute at Chicago (TTIC), for their early discussions and feedback to this project. ZW thank Zixin Wen at CMU for valuable suggestions in experiment design. JDL acknowledges support of the NSF CCF 2002272, NSF IIS 2107304, NSF CIF 2212262, ONR Young Investigator Award, and NSF CAREER Award 2144994.

## Impact Statement

This paper presents work whose goal is to advance the field of Machine Learning. There are many potential societal consequences of our work, none which we feel must be specifically highlighted here.

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

# A. Omitted proofs

## A.1. Theory Settings

To understand why a non-uniform data distribution helps learning compositional tasks, we aimed to find the simplistic setting where models the compositions of different skills. Inspired by the single-index model and parity tasks, we consider the following toy task with 'composing' $k$-scalar skills/functions.

**Setting** Fix skill number $d \in \mathbb{N}$ and integer 'composition' degree $k \geq 2$. Let $I_1, \ldots, I_k \overset{iid}{\sim} \{p_i\}_{i=1}^d$ on $[d]$. Let $\boldsymbol{x}_1, \ldots, \boldsymbol{x}_k \in \{\boldsymbol{e}_1, ..., \boldsymbol{e}_d\}$ be one-hot vectors with $\boldsymbol{x}_t = \boldsymbol{e}_{I_t} \in \mathbb{R}^d$. The training distribution is either uniform $\mathrm{Unif}([S])$ or $\mathrm{Zipf}(\alpha)$ distribution

$$\Pr[I_i = j] = p_j, \qquad p_j = \frac{j^{-a}}{H_{d,a}}, \quad H_{d,a} := \sum_{t=1}^d t^{-a}, \quad a > 1.$$

Here $\boldsymbol{x}_i$ are the 'name' of the functions and define the diagonal matrix $\boldsymbol{D} = \mathrm{diag}(p_1, \ldots, p_d)$ as the frequency that each function is sampled. We aim to show the difference between the training distributions.

**Model.** For $\boldsymbol{w} \in \mathbb{R}^d$, we define the function composition model as

$$f(\boldsymbol{w}, \boldsymbol{X}) = \prod_{i=1}^k (\boldsymbol{w}^\top \boldsymbol{x}_i), \qquad y = f(\boldsymbol{w}^\star, \boldsymbol{X}), \boldsymbol{X} = (\boldsymbol{x}_1, ..., \boldsymbol{x}_k)$$

where $\boldsymbol{w}^\star \in \mathbb{R}^d$ is fixed. Here, we see $f$ as the function composition as

$$g_{\boldsymbol{w}}(\boldsymbol{x}_k) \circ g_{\boldsymbol{w}}(\boldsymbol{x}_{k-1}) \circ \cdots \circ g_{\boldsymbol{w}}(\boldsymbol{x}_1)$$

where $g_{\boldsymbol{w}}(\boldsymbol{x}_i)(x) = (\boldsymbol{w}^\top \boldsymbol{x}_i)x$ with $x$ as the input scalar.

**Training objective.** We consider square loss as the training objective. The loss on a single sample $(\boldsymbol{X}^{(i)}, y^{(i)})$ is

$$\ell(\boldsymbol{w}; \boldsymbol{X}^{(i)}) = \frac{1}{2}\big(f(\boldsymbol{w}, \boldsymbol{X}^{(i)}) - y\big)^2, \boldsymbol{X}^{(i)} = \left(\boldsymbol{x}_1^{(i)}, ..., \boldsymbol{x}_k^{(i)}\right).$$

The population and empirical loss are

$$\mathcal{L}(\boldsymbol{w}) = \frac{1}{2} \mathbb{E}_{\boldsymbol{X}} \left[(f(\boldsymbol{w}, \boldsymbol{X}) - f(\boldsymbol{w}^\star, \boldsymbol{X}))^2\right], \qquad \hat{\mathcal{L}}(\boldsymbol{w}) = \frac{1}{n} \sum_{i=1}^n (f(\boldsymbol{w}, \boldsymbol{X}^{(i)}) - f(\boldsymbol{w}^\star, \boldsymbol{X}^{(i)}))^2.$$

## A.2. SQ lower bound under uniform inputs

In this section we can show that when the distribution is uniform, we need $d^{\Omega(k)}$ samples or exponential runtime $t \geq 2^{\Omega(d)}$ using a CSQ lower bound argument.

We consider the base function class $\mathcal{F} = \{f(\boldsymbol{w}, \cdot) : \boldsymbol{w} \in \{\pm 1\}^d\}$ with $\boldsymbol{w}$ on the unit hypercube. We define the inner product for each two functions $\boldsymbol{w}_1, \boldsymbol{w}_2$ as

$$\langle f_{\boldsymbol{w}_1}, f_{\boldsymbol{w}_2} \rangle = \mathbb{E}_{\boldsymbol{X}}[f(\boldsymbol{w}_1, \boldsymbol{X}) f(\boldsymbol{w}_2, \boldsymbol{X})] = \mathbb{E}_{\boldsymbol{X}} \prod_{i=1}^k (\boldsymbol{w}_1^\top \boldsymbol{x}_i \boldsymbol{x}_i^\top \boldsymbol{w}_2) = \left(\frac{\boldsymbol{w}_1^\top \boldsymbol{w}_2}{d}\right)^k$$

which satisfies $\mathbb{E}_{\boldsymbol{X}}[f(\boldsymbol{w}_1, \boldsymbol{X})^2] = 1$. Since we considered a square loss here, the gradient is

$$\nabla_{\boldsymbol{w}} \ell(\boldsymbol{X}) = (f(\boldsymbol{w}^*, \boldsymbol{X}) - f(\boldsymbol{w}, \boldsymbol{X})) \nabla_{\boldsymbol{w}} f(\boldsymbol{w}, \boldsymbol{X})$$

where the query satisfies the correlational query form $q(\boldsymbol{x}, y) = yf(\boldsymbol{x})$.

In order to prove the correlational statistical query lower bound, we first construct a function class $\mathcal{F}_k$ such that the inner product query provide little information about the target function. We can prove the following lemma by concentration.

**Lemma 1.** *There exists an absolute constant $c$ such that for any $\varepsilon > 0$, there exists a subset $S$ of $ce^{\frac{1}{4}\varepsilon^2 d}$ vectors on the hypercube $\{-1, 1\}^d$, such that for any $\boldsymbol{w}_1, \boldsymbol{w}_2 \in S$ with $\boldsymbol{w}_1 \neq \boldsymbol{w}_2$, we have*

$$\left| \frac{\boldsymbol{w}_1^\top \boldsymbol{w}_2}{d} \right| \leq \varepsilon.$$

*Proof.* Consider two random vectors $\boldsymbol{w}_1, \boldsymbol{w}_2$ sampling from the hypercube. By Hoeffding's Inequality, we have

$$\mathbb{P}\left[|\langle \boldsymbol{w}_1, \boldsymbol{w}_2 \rangle| \geq d\varepsilon\right] = \mathbb{P}\left[\left|\sum_{i=1}^{d} w_{1i} w_{2i}\right| \geq d\varepsilon\right] \leq 2\exp\left\{-\frac{1}{2}d\varepsilon^2\right\}.$$

By union bound, $N$ random rademacher vectors satisfy that for each pair of vectors has inner product $\leq d\varepsilon$ with probability

$$\mathbb{P}\left[|\langle \boldsymbol{w}_i, \boldsymbol{w}_j \rangle| \geq d\varepsilon, \forall i, j \in [N]\right] \leq \sum_{i<j} \mathbb{P}\left[|\langle \boldsymbol{w}_i, \boldsymbol{w}_j \rangle| \geq d\varepsilon\right] \leq 2N^2 \exp\left\{-\frac{1}{2}d\varepsilon^2\right\}.$$

We can pick $N = O(\exp(\frac{1}{4}d\epsilon^2))$ and the probability is less than 1, which means there exists such subset with the inner product of each pair of vectors satisfying $\left|\frac{\boldsymbol{w}_1^\top \boldsymbol{w}_2}{d}\right| \leq \varepsilon$. $\qquad\square$

Therefore, for any number of queries $q$, we can find $q$ unit vectors s.t. their pair-wise inner product are upper bounded by $d^{-1/2}\sqrt{\log q}$. Given the definition of the inner product, when $q = \text{poly}(d)$ we have

$$|\langle f_{\boldsymbol{w}_1}, f_{\boldsymbol{w}_2} \rangle| = (\boldsymbol{w}_1^\top \boldsymbol{w}_2) \lesssim d^{-k/2}.$$

Following standard CSQ arguments (Lemma 5 in Damian et al. (2022)), we directly have the theorem below. This is a restatement of Theorem 1 in the main paper.

**Theorem 3** (CSQ lower bound for uniform). *Let the input distribution be uniform $p_j = 1/d$ and $k \geq 2$. There exists a function class $\mathcal{F}_k$ and a constant $\epsilon = \Omega(1)$ such that any correlational statistical learner using $q$ queries requires a tolerance $\tau^2 \leq \left(\frac{\log(dq)}{d}\right)^{k/2}$ to achieve loss $\mathcal{L}(\boldsymbol{w}) \leq \epsilon$.*

Using the standard $\tau \approx \frac{1}{\sqrt{n}}$ concentration heuristic, where $n$ is the sample complexity, the theorem implies that either runtime is exponential in $d$ or sample size $n$ must be at least $\widetilde{\Omega}(d^{k/2})$, which is unacceptable when the number of skills $d$ is large and multiple hops of composition are required. Therefore, training under uniform distribution suffers from the computational gap and fails on such compositional task.

### A.3. Power law helps on sample complexity

The previous section has shown that uniform distribution hinders training when the compositional structure exists in the task by proving the statistical query lower bound. However, the lower bound only holds when the data distribution is uniform or symmetric. A power-law distribution is slightly different: the frequent skills occur with constant probability while tail skills are sampled much more infrequently. For simplicity, we consider $k$ is an even number with $k \geq 2$. The following theorem shows that gradient descent actually takes advantage of this asymmetry, which significantly improves the sample complexity.

**Theorem 4** (Gradient Descent learns to compose under power law training distribution). *Let the input distribution be Zipf law with $p_j \propto j^{-\alpha}$ with $\alpha > 1$. Suppose that the target error is $\varepsilon > 0$, and $\boldsymbol{w}(0) \sim \mathcal{N}(0_d, r^2 \boldsymbol{I}_d), r = \Theta(1), k = \Theta(1)$ is even, $\eta = O\left(\frac{1}{k^2\|\boldsymbol{p}\|_2}\right)$ and $B \geq \tilde{O}\left(\frac{\log\frac{1}{\delta}}{p_{\min}\varepsilon}\right)$. Then with probability $1 - \delta$ the model can learn the task by minibatch gradient descent with $\tilde{O}(\frac{d^{2\alpha}}{\eta\varepsilon})$ samples in $t \leq \tilde{O}(\frac{d^\alpha}{\eta}\log\frac{1}{\varepsilon})$ time, with error $\min\{\|\boldsymbol{w} + \boldsymbol{w}^*\|_\infty, \|\boldsymbol{w} - \boldsymbol{w}^*\|_\infty\} \leq \varepsilon$.*

*Proof.* The proof structure has two parts. We first prove the faster convergence when considering gradient descent on population loss, which characterizes the expected trajectory under the training distribution. Then, we do the finite sample analysis to track the sample complexity with online sampled minibatch SGD, which shows that the SGD trajectory

closely tracks the expected population trajectory based on concentration. In this phase, we show the convergence using a PL-condition like inequality, which still holds even with gradient noise.

First, we assume $\boldsymbol{w} = \mathbf{1}_d$ without loss of generality. This is because the gradient descent dynamics on $\boldsymbol{w}$ can be converted to an equivalent weight vector $\boldsymbol{u} := \boldsymbol{w} \odot \boldsymbol{w}^*$. Notice that $y^{(i)} = \prod_{t=1}^{k} \boldsymbol{w}^{*\top} \boldsymbol{x}_t^{(i)} \in \{-1, +1\}$. Thus the loss function for each sample becomes

$$\hat{\mathcal{L}}(\boldsymbol{w}) = \frac{1}{2} \sum_{i=1}^{n} \left( \prod_{t=1}^{k} \boldsymbol{w}^\top \boldsymbol{x}_t^{(i)} - \prod_{t=1}^{k} \boldsymbol{w}^{*\top} \boldsymbol{x}_t^{(i)} \right)^2 = \frac{1}{2} \sum_{i=1}^{n} \left( \prod_{t=1}^{k} \boldsymbol{u}^\top \boldsymbol{x}_t^{(i)} - 1 \right)^2$$

while the per-sample gradient is

$$\nabla \ell(\boldsymbol{w}) = \left( \prod_{t=1}^{k} \boldsymbol{u}^\top \boldsymbol{x}_t^{(i)} - 1 \right) \nabla_{\boldsymbol{w}} \prod_{t=1}^{k} \boldsymbol{u}^\top \boldsymbol{x}_t^{(i)} = \mathrm{diag}(\boldsymbol{w}^*) \nabla_{\boldsymbol{u}} \ell(\boldsymbol{w})$$

Therefore, based on gradient descent on $\boldsymbol{w}_t$, the update for $\boldsymbol{u}$ is

$$\boldsymbol{u}_{t+1} = \boldsymbol{w}^* \odot \boldsymbol{w}_{t+1} = \boldsymbol{u}_t - \eta \boldsymbol{w}^* \odot \nabla_{\boldsymbol{w}} \hat{\mathcal{L}} = \boldsymbol{u}_t - \eta \nabla_{\boldsymbol{u}} \hat{\mathcal{L}}.$$

This is equivalent to gradient descent on $\boldsymbol{u}$ directly and the ground truth is $\boldsymbol{u}^* = \mathbf{1}_d$.

After the reparametrization w.l.o.g., we start to analyze the population dynamics, which is the expected trajectory of the gradient descent dynamics. We denote the population iterates as $\boldsymbol{w}(t)$ and the empirical gradient descent iterates as $\hat{\boldsymbol{w}}(t)$.

We consider the two following quantities

$$A(t) = \langle \boldsymbol{w}(t), \boldsymbol{w}^* \rangle_p = \sum_{i=1}^{d} p_i w_i, \qquad B(t) = \|\boldsymbol{w}(t)\|_p^2 = \sum_{i=1}^{d} p_i w_i^2.$$

which are respectively the similarity between $\boldsymbol{w}$ and the ground truth and the norm under the power-law distribution.

The population gradient of $\boldsymbol{w}(t)$ can be written as

$$\nabla_{\boldsymbol{w}} \mathcal{L}(\boldsymbol{w}(t)) = \frac{1}{2} \nabla_{\boldsymbol{w}} \mathbb{E}_{\boldsymbol{X}} \left[ \left( \prod_{i=1}^{k} \boldsymbol{w}^\top \boldsymbol{x}_i - \prod_{i=1}^{k} \boldsymbol{w}^{*\top} \boldsymbol{x}_i \right)^2 \right]$$

$$= \frac{1}{2} \nabla_{\boldsymbol{w}} \mathbb{E}_{\boldsymbol{X}} \left[ \left( \prod_{i=1}^{k} \boldsymbol{w}^\top \boldsymbol{x}_i \right)^2 - 2 \left( \prod_{i=1}^{k} \boldsymbol{w}^\top \boldsymbol{x}_i \right) \left( \prod_{i=1}^{k} \boldsymbol{w}^{*\top} \boldsymbol{x}_i \right) + \left( \prod_{i=1}^{k} \boldsymbol{w}^{*\top} \boldsymbol{x}_i \right)^2 \right]$$

$$= \frac{1}{2} \nabla_{\boldsymbol{w}} \mathbb{E}_{\boldsymbol{X}} \left[ \left( \prod_{i=1}^{k} \boldsymbol{w}^\top \boldsymbol{x}_i \right)^2 - 2 \left( \prod_{i=1}^{k} \boldsymbol{w}^\top \boldsymbol{x}_i \right) \left( \prod_{i=1}^{k} \boldsymbol{w}^{*\top} \boldsymbol{x}_i \right) \right],$$

where the last step is because the last term is constant 1, whose gradient is zero.

By the independence of all $\boldsymbol{x}_i$s in one sample, we have

$$\left( \prod_{i=1}^{k} \boldsymbol{w}^\top \boldsymbol{x}_i \right)^2 = \|\boldsymbol{w}(t)\|_p^{2k} = B(t)^k, \qquad \left( \prod_{i=1}^{k} \boldsymbol{w}^\top \boldsymbol{x}_i \right) \left( \prod_{i=1}^{k} \boldsymbol{w}^{*\top} \boldsymbol{x}_i \right) = A(t)^k.$$

The population gradient can then be simplified to

$$\nabla_{\boldsymbol{w}} \mathcal{L}(\boldsymbol{w}(t)) = \nabla_{\boldsymbol{w}} \frac{1}{2} \left[ B(t)^k - 2A(t)^k \right] = k \boldsymbol{D} \left( B(t)^{k-1} \boldsymbol{w}(t) - A(t)^{k-1} \boldsymbol{w}^* \right)$$

We have the population gradient descent update:

$$\boldsymbol{w}_{t+1} = \boldsymbol{w}(t) - \eta k \boldsymbol{D} \left( B(t)^{k-1} \boldsymbol{w}(t) - A(t)^{k-1} \boldsymbol{w}^* \right)$$

By Lemma 3 and Lemma 4, we know the population landscape around initialization is "nice" enough for the PL condition on population loss to hold (Lemma 5),

$$\|\nabla\mathcal{L}(\boldsymbol{w}(t))\|_2^2 \geq 2kp_{\min}A(t)^{2k-2}\mathcal{L}(\boldsymbol{w}(t)),$$

which guarantees the gradient descent convergence on the population loss.

Finally, we consider the minibatch SGD trajectory and prove that it nearly follows the population dyanmics through finite sample analysis. We first define the gradient batch-noise. For each batch $B_t$ at time $t$, we have the batched gradient

$$\hat{g}_t = \frac{1}{B_t}\sum_{b_i=1}^{B_t}\nabla_{\boldsymbol{w}}\ell(\boldsymbol{w}(t), X_{b_i}).$$

whose expectation is the population gradient $\nabla\mathcal{L}(\boldsymbol{w}(t))$. We define the batch noise and sample noise for sample $X_{b_i}$ as

$$\xi_t = \hat{g}_t - \nabla\mathcal{L}(\boldsymbol{w}(t)), \qquad \xi_{t,i} = \nabla_{\boldsymbol{w}}\ell(\boldsymbol{w}(t), X_{b_i}) - \nabla\mathcal{L}(\boldsymbol{w}(t)).$$

We have the population loss decrement formula inductively by Lemma 8 (which means $\eta \leq \frac{1}{2L}$):

$$\mathcal{L}(\boldsymbol{w}_{t+1}) \leq \mathcal{L}(\boldsymbol{w}_t) - \eta\langle\nabla\mathcal{L}(\boldsymbol{w}(t)), \hat{g}_t\rangle + \frac{L\eta^2}{2}\|\hat{g}_t\|^2$$

$$= \mathcal{L}(\boldsymbol{w}(t)) - \eta\|\nabla\mathcal{L}(\boldsymbol{w}(t))\|^2 - \eta\langle\nabla\mathcal{L}(\boldsymbol{w}(t)), \xi_t\rangle + \frac{L\eta^2}{2}\left(\|\nabla\mathcal{L}(\boldsymbol{w}(t))\|^2 + 2\langle\nabla\mathcal{L}(\boldsymbol{w}(t)), \xi_t\rangle + \|\xi_t\|^2\right)$$

$$\leq \mathcal{L}(\boldsymbol{w}(t)) - \frac{\eta}{2}\|\nabla\mathcal{L}(\boldsymbol{w}(t))\|^2 + \frac{\eta}{2}|\langle\nabla\mathcal{L}(\boldsymbol{w}(t)), \xi_t\rangle| + \frac{\eta}{2}\|\xi_t\|^2.$$

With Lemma 8, we also have that with probability $1 - \frac{\delta t}{T}$, it holds that

$$\|\xi_t\| \leq \frac{1}{8}\|\nabla\mathcal{L}(\boldsymbol{w}(t))\|, \forall t \leq T.$$

So we have the following if we abbreviate $\mathcal{L}(\boldsymbol{w}(t)) := L_t$:

$$L_{t+1} \leq L_t - \frac{\eta}{2}\|\nabla L_t\|^2 + \frac{\eta}{2}\cdot(\frac{1}{8} + \frac{1}{64})\|\nabla L_t\|^2 \leq L_t - \frac{\eta}{3}\|\nabla L_t\|^2.$$

Given the PL-condition holds by the induction hypothesis Lemma 8, this will still lead to exponential decay of the population loss regardless of the gradient noise. By Lemma 8 we know with $T \geq \tilde{O}(\frac{1}{p_{\min}})$ the population loss $L_T \leq \epsilon$. By Lemma 6, we pick $\epsilon = O(p_{\min}\varepsilon^2)$ s.t. $\|\boldsymbol{w}(T) - \boldsymbol{w}^*\|_\infty \leq \varepsilon$, and the batch size $B \leq \tilde{O}\left(\frac{1}{\sqrt{p_{\min}\epsilon}}\right) = \tilde{O}(\frac{d^\alpha}{\varepsilon})$. This finishes the proof. $\square$

### A.3.1. POPULATION DYNAMICS TO CONVERGENCE

We prove the population dynamics converges to the ground truth vector $\boldsymbol{w}^*$ under the power-law distribution. The power-law distribution grants the initial similarity $A(0) = \Omega(1)$ with high probability, which boosts the initial alignment. With the initial alignments, we can show an improved landscape on the loss function with PL-style inequality. With those results, we can prove that GD pushes the iterates $\boldsymbol{w}(t)$ towards the ground truth.

Before the dynamics argument, we first prove that a landscape property of this task: there are only three possible stationary points for the loss function.

**Lemma 2.** *If the population gradient $\nabla_{\boldsymbol{w}}\mathcal{L}(\boldsymbol{w}(t)) = \boldsymbol{0}_d$, we have $\boldsymbol{w} = \boldsymbol{0}_d$ or $\boldsymbol{w} = \pm\boldsymbol{w}^*$.*

*Proof.* If $\nabla_{\boldsymbol{w}}\mathcal{L}(\boldsymbol{w}(t)) = \boldsymbol{0}_d$, we have $\left(B^{k-1}\boldsymbol{w} - A^{k-1}\boldsymbol{w}^*\right) = \boldsymbol{0}_d$. Therefore we know $\boldsymbol{w} = \left(\frac{A}{B}\right)^{k-1}\boldsymbol{w}^* := \lambda\boldsymbol{w}^*$. If $\lambda = 0$, the equality holds so 0 is one solution.

Now we consider $\lambda \neq 0$. Since $B = \|\boldsymbol{w}\|_p^2$ and $A = \langle\boldsymbol{w}, \boldsymbol{w}^*\rangle_p$, we can calculate the equation and have $\lambda = \frac{\lambda^{k-1}}{\lambda^{2k-2}}$. We have $\lambda^k = 1$, so $\lambda = \pm 1$. $\square$

This guarantees that gradient descent either drives $\boldsymbol{w}(t)$ to the only saddle point 0, or $\boldsymbol{w}(t)$ converges to the minimizer. We further prove that the power-law distribution guarantees that one will escape from the saddle point fast enough with random initialization, given stable learning rate.

**Stage 1: initialization** First, we use simple Gaussian concentration we have the following constant initial alignment between $w$ and $w^*$, which is the source of the separation between power law and the uniform distribution. Basically, power law or imbalanced training distribution significantly improved the training landscape.

**Lemma 3** (Gaussian lower bound for $|A(0)|$). *Assume $\boldsymbol{w}(0) \sim \mathcal{N}(0_d, r^2 \boldsymbol{I}_d)$. Then $A(0) = \sum_{i=1}^d p_i w_i(0)$ is Gaussian with $\mathrm{Var}(A(0)) = r^2 \sum_{i=1}^d p_i^2$. For any $\delta \in (0,1)$ and some constant $c_1, c_2$, with probability at least $0.999$, we have*

$$c_1 r \sqrt{\sum_{i=1}^d p_i^2} \leq |A(0)| \leq c_2 r \sqrt{\sum_{i=1}^d p_i^2}.$$

*Proof.* We have $\boldsymbol{w}(0)$ initialized as Gaussian, we have $A(0) = \sum_{i=1}^d p_i w_i(0)$. Since $A(0)$ is a linear functional of $\boldsymbol{w}(0)$, it follows that $A(0)$ is Gaussian. Moreover,

$$\mathbb{E}[A(0)] = p^\top \mathbb{E}[\boldsymbol{w}(0)] = 0,$$

and $\mathrm{Var}(A(0)) = \mathrm{Var}(p^\top \boldsymbol{w}(0)) = r^2 \|p\|_2^2 = r^2 \sum_{i=1}^d p_i^2$. Define $\sigma := r\|p\|_2$. Then $Z := A(0)/\sigma \sim \mathcal{N}(0,1)$. We now choose explicit absolute constants $c_1, c_2$ so that

$$\mathbb{P}\left(c_1 \leq |Z| \leq c_2\right) \geq 0.999.$$

Let $\delta_1 = \delta_2 = 5 \times 10^{-4}$. For the upper tail, for any $t \geq 0$,

$$\mathbb{P}(|Z| \geq t) = 2\Pr(Z \geq t) = 2\big(1 - \Phi(t)\big) \leq 2e^{-t^2/2},$$

where the last inequality is a standard Gaussian tail bound. If we set

$$c_2 := \sqrt{2\log\left(\frac{2}{\delta_2}\right)} = \sqrt{2\log(4000)},$$

then $\Pr(|Z| > c_2) \leq \delta_2$. For the lower tail, for any $t \geq 0$,

$$\Pr(|Z| \leq t) = \int_{-t}^t \frac{1}{\sqrt{2\pi}} e^{-x^2/2} \, dx \leq \int_{-t}^t \frac{1}{\sqrt{2\pi}} \, dx = \sqrt{\frac{2}{\pi}} \, t.$$

We set $c_1 := \delta_1 \sqrt{\frac{\pi}{2}}$, and $\mathbb{P}(|Z| < c_1) \leq \delta_1$. By union bound, we have

$$\mathbb{P}\left(c_1 \leq |Z| \leq c_2\right) \geq 1 - \Pr(|Z| < c_1) - \Pr(|Z| > c_2) \geq 1 - \delta_1 - \delta_2 = 0.999.$$

Multiplying the event $c_1 \leq |Z| \leq c_2$ by $\sigma = r\|p\|_2$ yields that w.p. at least $0.999$,

$$c_1 \, r\|p\|_2 \leq |A(0)| \leq c_2 \, r\|p\|_2,$$

which proves the claim. $\square$

Therefore, we have that the initial alignment is $\Theta(1)$ since $\|p\|_2 = \Theta(1)$. Similarly, we can prove the concentration for $B(0)$.

**Lemma 4** (Gaussian lower bound for $|B(0)|$). *Assume $\boldsymbol{w}(0) \sim \mathcal{N}(0_d, r^2 \boldsymbol{I}_d)$ and $B(0) = \sum_{i=1}^d p_i w_i^2(0)$. Then there exists an absolute constant $C > 0$ such that with probability at least $0.999$,*

$$\big|B(0) - r^2\big| \leq C r^2 \|p\|_2.$$

*Proof.* Write $w(0) = rg$ with $g \sim \mathcal{N}(0, I_d)$. Then

$$B(0) = r^2 g^\top \mathrm{diag}(p)\, g.$$

Since $\sum_i p_i = 1$, we have $\mathbb{E}[B(0)] = r^2 \mathrm{Tr}(\mathrm{diag}(p)) = r^2$. By the Hanson-Wright inequality for Gaussian quadratic forms (Rudelson & Vershynin, 2013),

$$\Pr\left(\big|g^\top A g - \mathrm{Tr}(A)\big| \geq t\right) \leq 2\exp\left(-c \min\left\{\frac{t^2}{\|A\|_F^2}, \frac{t}{\|A\|_2}\right\}\right)$$

for any symmetric matrix $A$. Applying this with $A = \text{diag}(p)$ yields

$$\Pr\left(\left|\sum_{i=1}^{d} p_i g_i^2 - 1\right| \geq t\right) \leq 2\exp\left(-c\min\left\{\frac{t^2}{\|p\|_2^2}, \frac{t}{\|p\|_2}\right\}\right).$$

Equivalently,

$$\Pr\left(|B(0) - r^2| \geq r^2 t\right) \leq 2\exp\left(-c\min\left\{\frac{t^2}{\|p\|_2^2}, \frac{t}{\|p\|_2}\right\}\right).$$

Now set $t = C\|p\|_2$, the exponent is bounded by $-c\min\{C^2, C\}$. Choosing $C$ sufficiently large makes the right-hand side at most $0.001$. Therefore, with probability at least $0.999$, $|B(0) - r^2| \leq Cr^2\|p\|_2$. This completes the proof. $\qquad\square$

After the analysis of the initialization scale, we know $|A(t)| > B(t)$ w.h.p. when $r \leq \frac{c_1\|p\|_2}{C\|p\|_2+1} = \Theta(1)$. We define the nice initialization event as $\mathcal{E}_0$. The population loss value is (since $k$ is even)

$$\mathcal{L}(\boldsymbol{w}(0)) = \frac{1}{2}\left(B^k(0) - 2A(0)^k + 1\right) \leq \frac{1}{2}(1 - A(0)^k).$$

**Stable learning rate analysis.** If gradient descent is in the stable regime ($\eta \leq \frac{1}{2\|\nabla^2\mathcal{L}(\boldsymbol{w})\|_2}$ for all time), we have descent lemma showing that $\mathcal{L}(\boldsymbol{w}(t))$ is non-increasing through time $t$. Then the monotonicity somehow implies the lower bound for $|A(t)|$:

$$\frac{1}{2}\left(1 - 2A(t)^k\right) \leq \frac{1}{2}\left(B^k(t) - 2A(t)^k + 1\right) = \mathcal{L}(\boldsymbol{w}(t)) \leq \mathcal{L}(\boldsymbol{w}(0)) \leq \frac{1}{2}(1 - A(0)^k).$$

Therefore $|A(t)| \geq 2^{-1/k}|A(0)|$, $B^k(t) \leq 2A(t)^k$ as long as GD trajectory is stable.

On the other hand, we can upper bound smoothness constant $L$ to estimate the max possible learning rate along the training trajectory. Given the population gradient

$$\nabla_{\boldsymbol{w}}\mathcal{L}(\boldsymbol{w}(t)) = k\boldsymbol{D}\left(B(t)^{k-1}\boldsymbol{w}(t) - A(t)^{k-1}\boldsymbol{w}^*\right),$$

we can calculate the Hessian matrix

$$\nabla^2\mathcal{L}(\boldsymbol{w}(t)) = kB(t)^{k-1}\boldsymbol{D} + 2k(k-1)B(t)^{k-2}\boldsymbol{Dw}(t)(\boldsymbol{Dw}(t))^\top - k(k-1)A(t)^{k-2}\boldsymbol{pp}^\top.$$

To upper bound the operator norm of the Hessian, we use

$$\|\boldsymbol{D}\|_{op} = p_{\max}, \quad \left\|\boldsymbol{Dw}(t)(\boldsymbol{Dw}(t))^\top\right\|_{op} = \sum_{i=1}^{d} p_i^2 w_i(t)^2 \leq p_{\max}B(t), \quad \left\|\boldsymbol{pp}^\top\right\|_{op} \leq \|\boldsymbol{p}\|_2^2.$$

Therefore, we have (using $B^k(t) \leq 2|A(t)|^k$)

$$\left\|\nabla^2\mathcal{L}(\boldsymbol{w}(t))\right\|_{op} \leq kB(t)^{k-1}p_{\max} + 2k(k-1)B(t)^{k-1}p_{\max} + k(k-1)A(t)^{k-2}\|\boldsymbol{p}\|_2^2$$
$$\leq 2k(2k-1)p_{\max}|A(t)|^{k-1} + k(k-1)\|\boldsymbol{p}\|_2^2|A(t)|^{k-2}.$$

Since we know $B(t) = \left(\sum_{i=1}^{d} p_i w_i(t)^2\right) \cdot \left(\sum_{i=1}^{d} p_i\right) \geq A(t)^2$ by Cauchy, we can further upper bound

$$A(t)^{2k} \leq 2A(t)^k, |A(t)| \leq 2^{1/k}, |B(t)| \leq 2^{2/k}.$$

That gives $\left\|\nabla^2\mathcal{L}(\boldsymbol{w}(t))\right\|_{op} \leq 3k^2\|\boldsymbol{p}\|_2$ which is a constant upper bound. We pick $\eta \leq \frac{1}{10k^2\|\boldsymbol{p}\|_2}$ and that will guarantee stable training since the first iterate (as induction base case, the upper bound hold trivially at $t = 0$).

**Stage 2: convergence**  With the lower bounds of $A(t)$, we can use the Polyak–Łojasiewicz condition on the population loss to calculate the time for convergence to the global minima (either $\boldsymbol{w}^*$ or $-\boldsymbol{w}^*$).

**Lemma 5** (PL inequalities). *When $r \leq \frac{c_1 \|\boldsymbol{p}\|}{C\|\boldsymbol{p}\|+1}, \eta \leq \frac{1}{10k^2\|\boldsymbol{p}\|_2}$, we have*

$$\|\nabla\mathcal{L}(\boldsymbol{w}(t))\|_2^2 \geq 2kp_{\min}A(t)^{2k-2}\mathcal{L}(\boldsymbol{w}(t)).$$

*Proof.*  Consider the L.H.S. Since $\lambda_{\min}(D) = p_{\min}$, we have:

$$\begin{aligned}
\|\nabla\mathcal{L}(\boldsymbol{w}(t))\|_2^2 &= k^2\big(B(t)^{k-1}\boldsymbol{w}(t) - A(t)^{k-1}\boldsymbol{w}^*\big)^\top \boldsymbol{D}^2\big(B(t)^{k-1}\boldsymbol{w}(t) - A(t)^{k-1}\boldsymbol{w}^*\big) \\
&\geq k^2 p_{\min}\big(B(t)^{k-1}\boldsymbol{w}(t) - A(t)^{k-1}\boldsymbol{w}^*\big)^\top \boldsymbol{D}\big(B(t)^{k-1}\boldsymbol{w}(t) - A(t)^{k-1}\boldsymbol{w}^*\big) \\
&= k^2 p_{\min}\big(B(t)^{2k-1} - 2A(t)^k B(t)^{k-1} + A(t)^{2k-2}\big)
\end{aligned}$$

We next prove that

$$k\big(B(t)^{2k-1} - 2A(t)^k B(t)^{k-1} + A(t)^{2k-2}\big) \geq 2A^{2k-2}\mathcal{L}(\boldsymbol{w}(t)).$$

We know that $|A(t)|^2 \leq B(t)$ and

$$2\mathcal{L}(\boldsymbol{w}(t)) = B(t)^k - 2A(t)^k + 1 = A(t)^{2k}\left(\frac{B}{A^2}\right)^k - 2A^k + 1.$$

Let $A^k = a, \frac{B}{A^2} = b$. Then the both side can be rewritten into

$$k\big(B(t)^{2k-1} - 2A(t)^k B(t)^{k-1} + A(t)^{2k-2}\big)/A^{2k-2} = ka^2 b^{2k-1} - 2kab^{k-1} + k,$$

$$2\mathcal{L}(\boldsymbol{w}(t)) = a^2 b^k - 2a + 1.$$

We need to prove that $(kb^{k-1} - 1)(b^k a^2 - 2a) + k - 1 \geq 0$. Since $b \geq 1$, we know

$$\begin{aligned}
(kb^{k-1} - 1)(b^k a^2 - 2a) + k - 1 &= (kb^{k-1} - 1)(b^k a^2 - 2a + \frac{1}{b^k}) + k - 1 - \frac{1}{b^k}(kb^{k-1} - 1) \\
&\geq k - 1 - \frac{1}{b^k}(kb^{k-1} - 1) = \frac{kb^k - kb^{k-1} + 1}{b^k} - 1 \geq 0
\end{aligned}$$

when $b \geq 1$ since the function is increasing after $b \geq 1$. Therefore we finish the proof.  □

With the PL condition we can easily prove the convergence of the loss. By descent lemma (since we picked $\eta \leq \frac{1}{|\lambda_{\max}(\nabla^2\mathcal{L})|}$), we have

$$\mathcal{L}(\boldsymbol{w}(t+1)) \leq \mathcal{L}(\boldsymbol{w}(t)) - \frac{\eta}{2}\|\nabla\mathcal{L}(\boldsymbol{w}(t))\|_2^2 \leq \left(1 - \frac{\eta k p_{\min} A(t)^{2k-2}}{2}\right)\mathcal{L}(\boldsymbol{w}(t)).$$

Since $|A(t)| \geq 2^{-1/k}|A(0)|$, we have

$$\frac{\eta k p_{\min} A(t)^{2k-2}}{2} \geq \frac{\eta k p_{\min} 2^{-2+2/k}|A(0)|^{2k-2}}{2}$$

The loss converges to $\epsilon_1$ within $t \leq O(\frac{1}{\eta k p_{\min}}\log\frac{1}{\epsilon_1})$ where $\epsilon_1$ will be determined later.

Finally, $\boldsymbol{w}(t)$ will be very close to the ground truth if the population loss is very small.

**Lemma 6.** *Assume the 'good' initialization holds. When $\mathcal{L}(\boldsymbol{w}) \leq \epsilon_1 \leq \frac{1}{8}$, we have $\min\{\|\boldsymbol{w} + \boldsymbol{w}^*\|_\infty, \|\boldsymbol{w} - \boldsymbol{w}^*\|_\infty\} \leq O(\sqrt{\frac{\epsilon_1}{p_{\min}}})$.*

*Proof.*  W.l.o.g. we assume $A(0) > 0$, which guarantees that $A(t) > 0$ for all $t$.

We have $\mathcal{L}(\boldsymbol{w}(0)) \geq \mathcal{L}(\boldsymbol{w}(t)) \geq \frac{1}{2}\left(1 - 2A(t)^k + B(t)^k\right) \geq \frac{1}{2}\left(1 - A(t)^k\right)^2$ because $B(t) \geq A(t)^2$. That means $\left(A(t)^k - 1\right)^2 \leq 2\epsilon_1$. We also have $A(t)^k \leq 2$ and $|A(t)| \geq 2^{-1/k}|A(0)|$, so

$$\sqrt{2\epsilon_1} \geq \left|A^k(t) - 1\right| = \left|(A(t) - 1)\sum_{i=0}^{k-1} A(t)^i\right| \geq |A(t) - 1|.$$

That leads to $(A(t) - 1)^2 \leq 2\epsilon_1$.

We can similarly upper bound $B - A^2$ : by $A \leq 2^{1/k}, B \geq A^2$ and $B \leq 2^{2/k}$, we have

$$2\mathcal{L}(\boldsymbol{w}(t)) \geq B^k - A^{2k} = (B - A^2)\sum_{j=0}^{k-1} B^{k-1-j}A^{2j} \geq \sum_{j=0}^{k-1} A^{2k-2}(B - A^2) \geq kA(0)^{2k-2}(B - A^2).$$

So $B - A^2 \leq \frac{2\epsilon_1}{kA(0)^{2k-2}}$. Combine both terms, we have

$$(B - A^2) + (A - 1)^2 = \sum_{i=1}^{d} p_i(w_i - w_i^*)^2 \leq O(\epsilon_1).$$

which gives $\|\boldsymbol{w} - \boldsymbol{w}^*\|_\infty \leq O\left(\sqrt{\frac{\epsilon_1}{p_{\min}}}\right)$. The proof is exactly the same when $A(0) < 0$. $\qquad\square$

### A.3.2. FINITE SAMPLE ANALYSIS

Finally, we apply minibatch stochastic gradient descent and prove that the descent trajectory follows the population dynamics. During the finite sample analysis, we need to inductively prove that the batch noise each step is bounded for all time, and the PL-condition always holds on the population dynamics. In this way, we still ensure the population loss decrement by the minibatch gradient descent updates.

Recall the definition of the gradient batch-noise. For each batch $B_t$ at time $t$, we have the batched gradient

$$\hat{g}_t = \frac{1}{B_t}\sum_{b_i=1}^{B_t} \nabla_{\boldsymbol{w}}\ell(\boldsymbol{w}(t), X_{b_i}).$$

whose expectation is the population gradient $\nabla\mathcal{L}(\boldsymbol{w}(t))$. Since the batch noise and sample noise for the sample $X_{b_i}$ are

$$\xi_t = \hat{g}_t - \nabla\mathcal{L}(\boldsymbol{w}(t)), \xi_{t,i} = \nabla_{\boldsymbol{w}}\ell(\boldsymbol{w}(t), X_{b_i}) - \nabla\mathcal{L}(\boldsymbol{w}(t)).$$

Assume that the smoothness constant has the upper bound $L$ (which we will upper bound inductively). Then we have the population loss decrement formula (we pick $\eta \leq \frac{1}{2L}$):

$$
\begin{aligned}
\mathcal{L}(\boldsymbol{w}_{t+1}) &\leq \mathcal{L}(\boldsymbol{w}_t) - \eta\langle\nabla\mathcal{L}(\boldsymbol{w}(t)), \hat{g}_t\rangle + \frac{L\eta^2}{2}\|\hat{g}_t\| \\
&= \mathcal{L}(\boldsymbol{w}(t)) - \eta\|\nabla\mathcal{L}(\boldsymbol{w}(t))\|^2 - \eta\langle\nabla\mathcal{L}(\boldsymbol{w}(t)), \xi_t\rangle + \frac{L\eta^2}{2}\left(\|\nabla\mathcal{L}(\boldsymbol{w}(t))\|^2 + 2\langle\nabla\mathcal{L}(\boldsymbol{w}(t)), \xi_t\rangle + \|\xi_t\|^2\right) \\
&\leq \mathcal{L}(\boldsymbol{w}(t)) - \frac{\eta}{2}\|\nabla\mathcal{L}(\boldsymbol{w}(t))\|^2 + \frac{\eta}{2}|\langle\nabla\mathcal{L}(\boldsymbol{w}(t)), \xi_t\rangle| + \frac{\eta}{2}\|\xi_t\|^2.
\end{aligned}
$$

Therefore, we just need to control the failure probability that $\|\xi_t\|$ exceeds a certain threshold. We note that if

$$\|\xi_t\| \leq \frac{1}{8}\|\nabla\mathcal{L}(\boldsymbol{w}(t))\|,$$

we will have (recall that we abbreviate $\mathcal{L}(\boldsymbol{w}(t)) = L_t$):

$$L_{t+1} \leq L_t - \frac{\eta}{2}\|\nabla L_t\|^2 + \frac{\eta}{2} \cdot \left(\frac{1}{8} + \frac{1}{64}\right)\|\nabla L_t\|^2 \leq L_t - \frac{\eta}{3}\|\nabla L_t\|^2.$$

And if the PL-condition holds, that will still lead to exponential decay of the population loss regardless of the gradient noise. Next, we need to prove that the events $\|\xi_t\| \leq \frac{1}{8}\|\nabla \mathcal{L}(\boldsymbol{w}(t))\|$ happen with very high probability inductively for all $t$, together with the boundedness conditions it requires.

We now write down the concentration inequalities that bounded the norm of the gradient noise given the uniform upper bound of the parameter $\|\boldsymbol{w}(t)\|_\infty$.

**Lemma 7.** *If $\|\boldsymbol{w}(t)\|_\infty \leq R$ for all $t$, there exist some absolute constant $c_3, c_4$ s.t. for all $t > 0$:*

$$\mathbb{P}\left[\|\xi_t\| \geq s\right] \leq 2\exp\left(-\frac{Bs^2}{c_3 k^2 R^{2k-2} L_t + c_4 k R^{2k-1} s}\right).$$

*In particular, when $s = \frac{1}{8}\|\nabla L_t\|$ and $B \geq O\left(k^2 \max\{R^{2k-2}, R^{2k-1}\}\left(\frac{L_t}{s^2} + \frac{1}{s}\right)\log\frac{2}{\delta_t}\right)$, we have with probability $1 - \delta_t$, $\|\xi_t\| \leq s$.*

*Proof.* We first upper bound $\|\xi_{t,i}\|$ uniformly, which requires the upper bound for the sampled gradient $\nabla\ell_t(X)$:

$$\nabla\ell_t(X) = \left(\prod_{i=1}^{k}(\boldsymbol{w}^\top \boldsymbol{x}_i) - \prod_{i=1}^{k}\boldsymbol{w}^{*\top}\boldsymbol{x}_i\right)\nabla\sum_{i=1}^{k}\prod_{j\neq i}(\boldsymbol{w}^\top \boldsymbol{x}_j)\boldsymbol{x}_i \leq 2kR^{2k-1}.$$

Therefore the batch expectation and population gradient expectation can both be upper bounded by this, and we have

$$\|\xi_{t,i}\| \leq \|\hat{g}_{t,i}\| + \|\nabla L_t\| \leq 4kR^{2k-1}.$$

Then we calculate the second moment of $\|\xi_{t,i}\|$. We know $\mathbb{E}[\|\xi_{t,i}\|^2] \leq \mathbb{E}_X[\|\hat{g}_{t,i}\|^2]$. While

$$\|\hat{g}_{t,i}\|^2 \leq (f(\boldsymbol{w}, X_{b_i}) - 1)^2\|\nabla f\|^2 \leq (f(\boldsymbol{w}, X_{b_i}) - 1)^2 k^2 R^{2k-2}.$$

So $\mathbb{E}[\|\xi_{t,i}\|^2] \leq 2k^2 R^{2k-2}L_t$. Finally, we apply vector Bernstein inequality in Hilbert Space (Martinez-Taboada & Ramdas (2024), Appendix D.3), and we finish the proof. $\square$

With the concentration inequality, we can formulate our induction. We show that if all the high probability event happens, including the initialization event $\mathcal{E}_0$ and all the batch gradient noise concentration, we can show (1) the uniform upper bound for $\|\boldsymbol{w}\|_\infty$ (2) monotonic decrement on the population loss. With the induction lemma, a final union bound finishes the proof of the main theorem.

**Lemma 8** (Induction). *If $k = \Theta(1), k \geq 2, \eta \leq O\left(\frac{1}{k^2\|\boldsymbol{p}\|_2}\right)$ and $B \geq \tilde{O}\left(\frac{\log\frac{1}{\delta}}{\sqrt{p_{\min}}\epsilon}\right)$, we show that with probability $1 - \delta$, the following holds for all $t \geq t_0 + 1$ if they are satisfied with $t \leq t_0$:*

1. $\|\boldsymbol{w}(t)\|_\infty \leq R := 2 + \left(\frac{A(0)}{2}\right)^{-\frac{k-1}{k}}$.

2. $L_{t+1} \leq \left(1 - \frac{\eta k p_{\min}|A(0)|^{2k-2}}{6}\right)L_t$ when $L_t \geq \epsilon = O(p_{\min})$.

*The induction base is also satisfied when $t = 0$. Finally, $L_T \leq \epsilon$ with $T \leq \tilde{O}\left(\frac{1}{\eta p_{\min}}\log\frac{1}{\epsilon}\right)$.*

*Proof.* We first prove the $t = 0$ case. The first requirement is satisfied when the initialization is nice, i.e. when $\mathcal{E}_0$ holds with failure probability $\delta/(T+1)$. Then we prove the loss decrement. For the first step gradient, we know $\|\nabla L_0\| \geq \Theta(1)$. By the previous concentration Lemma 7, $\|\xi_0\| \leq \frac{1}{8}\|\nabla L_0\|$ holds with failure probability $\delta/(T+1)$. Thus, we have

$$L_1 \leq L_0 - \frac{\eta}{2}\|\nabla L_0\|^2 + \frac{\eta}{2}\cdot\left(\frac{1}{8} + \frac{1}{64}\right)\|\nabla L_0\|^2 \leq L_0 - \frac{\eta}{3}\|\nabla L_0\|^2.$$

With the PL-condition $\|\nabla L_0\|_2^2 \geq 2k p_{\min}A(0)^{2k-2}\mathcal{L}(\boldsymbol{w}(0))$, we know the second hypothesis for base case holds.

Now we assume for all $t' \leq t$, the induction holds with failure probability $\frac{t\delta}{T+1}$. First, we upper bound the population gradient norm by the population loss (since $B(t) \geq A^2(t)$):

$$\begin{aligned}
\|\nabla\mathcal{L}(\boldsymbol{w}(t))\|_2^2 &= k^2\big(B(t)^{k-1}\boldsymbol{w}(t) - A(t)^{k-1}\boldsymbol{w}^*\big)^\top \boldsymbol{D}^2\big(B(t)^{k-1}\boldsymbol{w}(t) - A(t)^{k-1}\boldsymbol{w}^*\big) \\
&\leq k^2 p_{\max}\big(B(t)^{k-1}\boldsymbol{w}(t) - A(t)^{k-1}\boldsymbol{w}^*\big)^\top \boldsymbol{D}\big(B(t)^{k-1}\boldsymbol{w}(t) - A(t)^{k-1}\boldsymbol{w}^*\big) \\
&= k^2\big(B(t)^{2k-1} - 2A(t)^k B(t)^{k-1} + A(t)^{2k-2}\big) \\
&= k^2 A^{2k-2}\Big(B^k\big(\frac{B^{k-1}}{A^{2k-2}}\big) - 2A(t)^k\big(\frac{B^{k-1}}{A^{2k-2}}\big) + 1\Big) \leq 2k^2 B^{k-1} L_t.
\end{aligned}$$

By the induction on the decreasing population loss, we still have $|A| \leq 2^{1/k}, |B| \leq 2^{2/k}$. Also, $|A(t)|$ has the uniform lower bound $|A(t)| \geq \frac{1}{2^{1/k}}|A(0)|$.

Now we prove that the batch size is large enough for the concentration. When we pick $s = \frac{1}{8}\|\nabla L_t\|$, the necessary batch size is ($k = \Theta(1)$, we abbreviate all constants.)

$$B_t = O\Big(k^2 R^{2k-2}\Big(\frac{L_t}{s^2} + \frac{1}{s}\Big)\log\frac{2}{\delta_t}\Big) = O\Big(\Big(\frac{L_t}{s^2} + \frac{1}{s}\Big)\log\frac{2}{\delta_t}\Big).$$

Note that we have the PL-condition $\|\nabla\mathcal{L}(\boldsymbol{w}(t))\|_2^2 \geq 2k p_{\min} A(t)^{2k-2}\mathcal{L}(\boldsymbol{w}(t))$ and $|A(t)| \geq \frac{1}{2^{1/k}}|A(0)| = \Theta(1)$. Therefore the coefficient $\big(\frac{L_t}{s^2} + \frac{1}{s}\big)$ has the upper bound

$$\Big(\frac{L_t}{s^2} + \frac{1}{s}\Big) \leq O\Big(\frac{L_t}{p_{\min} L_t} + \frac{1}{\sqrt{p_{\min} L_t}}\Big) \leq O\Big(\max\Big\{\frac{1}{p_{\min}}, \frac{1}{\sqrt{p_{\min}\epsilon}}\Big\}\Big).$$

That means $B \geq \tilde{O}(\frac{1}{\sqrt{p_{\min}\epsilon}})$ suffices for the concentration with $\epsilon \leq O(p_{\min})$.

We then prove that it will not exceed the norm upper bound given the concentration inequality on this batch holds. We directly calculate the gradient for each entry $w_i(t)$ :

$$\nabla\mathcal{L}(\boldsymbol{w})_i = k p_i(B(t)^{k-1}w_i(t) - A(t)^{k-1}w_i^*), \nabla\hat{\mathcal{L}}_{B_t}(\boldsymbol{w})_i = \nabla\mathcal{L}(\boldsymbol{w})_i + \xi_{t,i}$$

If $w_i \leq R-1$, we know one step gradient descent with learning rate $\eta$ won't exceed the limit since the batched gradient norm is upper bounded:

$$|w_i(t+1)| \leq |w_i(t)| + \eta|\nabla\hat{\mathcal{L}}_{B_t}(\boldsymbol{w})_i| \leq R-1 + 2\eta k^2 B^{k-1} L_t \leq R.$$

If $w_i \geq R-1$, the population gradient will actually point to the shrinking direction.

$$\begin{aligned}
\nabla\mathcal{L}(\boldsymbol{w})_i &= k p_i(B(t)^{k-1}w_i(t) - A(t)^{k-1}w_i^*) \\
&= k p_i B^{k-1}(w_i(t) - \frac{A(t)^{k-1}}{B(t)^{k-1}}w_i^*) \\
&\geq k p_i B^{k-1}\Big(w_i(t) - \frac{1}{|A(t)|^{k-1}}\Big) \geq k p_i B^{k-1}.
\end{aligned}$$

The last two inequalities are because $A(t)^2 \leq B(t)$, the upper bound for $|w_i|$ and the uniform lower bound for $|A(t)|$. With the concentration, the norm of noise is upper bounded by the population gradient, so the coordinate $w_i$ will shrink towards 0 and guarantee that $\|\boldsymbol{w}(t+1)\|_\infty \leq R$. The first argument thus holds for time $t+1$.

Next we prove the loss decrement argument. We have the descent formula when the concentration inequalities $\|\xi_t\| \leq \frac{1}{8}\|\nabla L_t\|$ hold:

$$L_{t+1} \leq L_t - \frac{\eta}{2}\|\nabla L_t\|^2 + \frac{\eta}{2}\cdot(\frac{1}{8} + \frac{1}{64})\|\nabla L_t\|^2 \leq L_t - \frac{\eta}{3}\|\nabla L_t\|^2.$$

Use the PL-condition $\|\nabla\mathcal{L}(\boldsymbol{w}(t))\|_2^2 \geq 2k p_{\min} A(t)^{2k-2}\mathcal{L}(\boldsymbol{w}(t))$ and $|A(t)| \geq \frac{1}{2^{1/k}}|A(0)|$. Therefore we have

$$L_{t+1} \leq (1 - \frac{2\eta k p_{\min} 2^{-2+2/k}|A(0)|^{2k-2}}{3})L_t \leq (1 - \frac{\eta k p_{\min}|A(0)|^{2k-2}}{6})L_t.$$

With time $T \geq O\Big(\frac{1}{\eta k p_{\min}|A(0)|^{2k-2}}\log\frac{L_0}{\epsilon}\Big) = \tilde{O}(\frac{1}{p_{\min}})$, the population loss $L_T \leq \epsilon$. $\qquad\square$

We finally apply Lemma 6 and have $\min\{\|\boldsymbol{w}(T) + \boldsymbol{w}^*\|_\infty, \|\boldsymbol{w}(T) - \boldsymbol{w}^*\|_\infty\} \leq O\left(\sqrt{\frac{\epsilon}{p_{\min}}}\right)$. We pick $\epsilon = O\left(p_{\min}\varepsilon^2\right)$. For any $\varepsilon > 0$, we need in total

$$N = B \cdot T = \tilde{O}\left(\frac{1}{\sqrt{p_{\min}^2 \varepsilon^2}} \cdot \frac{1}{p_{\min}}\right) = \tilde{O}\left(\frac{d^{2\alpha}}{\varepsilon}\right)$$

samples to make $\min\{\|\boldsymbol{w}(T) + \boldsymbol{w}^*\|_\infty, \|\boldsymbol{w}(T) - \boldsymbol{w}^*\|_\infty\} \leq \varepsilon$. Thus we finish the proof.

**Remark on the sample complexity** . The sample complexity depends on $k$ even though it does not appear on the exponent of $d$. Since $k$ is $\Theta(1)$ and $d \gg 1$, we write the sample complexity in the form of a polynomial of $d$. For simplicity and clarity, we hide the constant terms with $k$ in the $O(\cdot)$ notation. If $k$ is also taken into account, the sample complexity should be $O(c^k d^{2\alpha})$ v.s. $d^{\Omega(k)}$ where $c = \Theta(1) \ll d$ is a constant that only depends on $\alpha$. Intuitively, this is a necessary price to pay to escape a $k$-th order saddle. That means the sample complexity for the power law also increases with $k$, but the rate is much slower. Similar sample complexity separations can be found in the learning multi-index model literature like Damian et al. (2022).

## B. Multi-step Arithmetic

We use a basic multi-step arithmetic task with operators $(+, -, \times)$, 4 operators, and operands sampled uniformly from $[1, 50]$. We provide the formula expression to the model and ask it to directly output the answer. The expressions follow standard mathematical precedence rules where multiplication is evaluated before addition and subtraction. We use the Qwen3 tokenizer (Yang et al., 2025) to tokenize the prompts and labels.

During training, we use the AdamW optimizer with $(\beta_1, \beta_2) = (0.9, 0.999)$ and weight decay $0.01$. We train a Qwen3-0.6B model from scratch (random initialization) for $10{,}000$ steps with a per-device batch size of $128$ across 8 gpus. We use a peak learning rate of $3 \times 10^{-4}$ with cosine decay and $500$ warmup steps. The model is trained in bfloat16 precision.

For the Zipf distribution experiments, we first randomly shuffle the integers in $[1, 50]$ to obtain a permutation $\pi$, then sample operands according to a power-law distribution where the probability of sampling the $k$-th element $\pi(k)$ is:

$$p_k \propto \frac{1}{k^\alpha}, \quad k = 1, 2, \ldots, 50 \tag{1}$$

with Zipf exponent $\alpha = 1.0$.

We evaluate every $500$ steps on $100{,}000$ uniformly sampled test expressions with a fixed random seed to ensure consistency. The prompt format is:

> **Example: Multi-step Arithmetic**
>
> **Prompt:** "User: Calculate 23 + 15 * 7 - 42 * 3.\nAssistant:\boxed{"
> **Label:** " 2}"

where we prefill "\boxed{" and train the model to generate only the answer followed by "}". Note that the Qwen3 tokenizer treats "{-" as a single token, so we add a space before the answer to ensure consistent tokenization between training and evaluation for negative numbers.

## C. State tracking on $S_5$

We use the experiment setting close to the **Permutation Composition** task in Li et al. (2024b). We only consider the permutation group $S_5$, which is the smallest unsolvable symmetry group. For each permutation, it takes 5 tokens to represent. The target is the composed permutation. We only consider 4-hop state tracking composition task. The vocabulary size is only 5 with $\{1, 2, 3, 4, 5\}$. We directly train the embedding layer in this setting. The input sequence length is $5 \times 4 = 20$. The loss is only calculated on the last 5 token positions for predicting the final composition.

During training, we use the AdamW optimizer with $(\beta_1, \beta_2) = (0.9, 0.999), \epsilon = 10^{-8}$ and weight decay $10^{-6}$. We train an encoder transformer with 4 layers, 256 hidden dimension (random initialization) for 200k steps with a gpu of batch size of 256. All the data is generated on the fly. The peak learning rate is $2 \times 10^{-4}$ with cosine decay to $0.1\times$ of peak and 1000

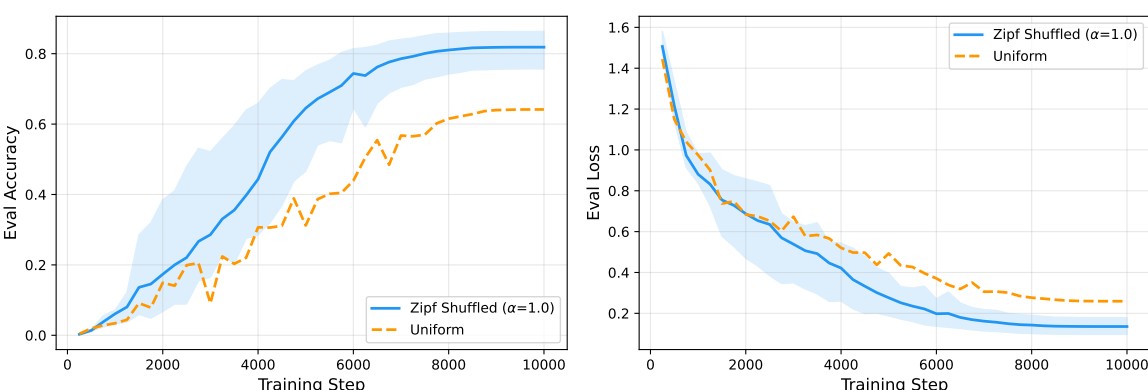

*Figure 6.* Results of Arithmetic from scratch (Qwen3-0.6B, 4 ops over $[1, 50]$, operators $\{+, -, \times\}$). Zipf($\alpha$=1.0) with shuffled rank-to-value mapping vs uniform sampling, evaluated on uniformly sampled test expressions. Solid line represents the mean over 4 seeds; shaded region represents the min-max range. Both the eval set and the loss validation set are sampled from *uniform* distribution. Zipf reaches $\sim$82% accuracy vs $\sim$64% for uniform training set.

warmup steps. The model is trained with fp16. By default we use lexicographical order, and $\alpha = 1.0$. The experiment in the main figure is $\alpha = 1.5$ with random order, in order to ablate the effect of lexicographical order and only exhibit the effect of the asymmetry. All test dataset is using uniform distribution if without specification.

For understanding the learning order of the different skills, we further include five different skill bins (rank 0-20%, ..., 80%-100%) to measure the learning accuracy/loss. In the appendix, the average (k-hop) loss/accuracy curves are online test loss/accuracy, depending the training power law loss.

> **Example: State Tracking ($S_5$)**
>
> **Input:** The $k$ permutations: (**1 2 3 4 5** 1 2 3 4 5 **1 3 2 4 5** 1 2 3 4 5).
> **Output:** (1 3 2 4 5).

### C.1. The effect of the exponent $\alpha$

In this section, we ablate the effect of the exponent of $\alpha$. We train five different $\alpha \in \{0.5, 0.75, 1.0, 1.25, 1.5\}$ with the same initialization and fix the order of the permutations as lexicographical order. The results are as shown in Figure 7 (failed runs with small alpha 0.5 and 0.75) and Figure 8 (success runs with larger alpha 1.0, 1.25 and 1.5). The result actually echoes our theory and indicates that there is a trade-off on the exponent $\alpha$.

The summary of the results are:

- **Large enough $\alpha$ is necessary**. We find that when $\alpha$ is too small such as 0.5, the optimization still fails or gets stuck in an early phase, as shown in Figure 7. That corresponds to the sufficient condition that $\alpha > 1$ in our theorem. Though the condition is not a necessary condition for $k$-multiplicative composition nor provably transferrable to $S_5$ composition tasks, the intuition holds that **if the head probability is not large enough**, the optimization landscape won't good enough for learning.

- **Larger $\alpha$ leads to faster initial descent, but suffers more with long-tail**: As shown in Figure 8, **larger $\alpha$** leads to **faster training at first**. The test loss in Bin 1 quickly decreases and enable better learning of the compositional skill generally (e.g. $\alpha = 1.5$ converge much faster than $\alpha = 1.0$ or 1.25). However, the tail skills' learning will be slowed down due to smaller sampling probability, so the $\alpha = 1.5$ one falls behind in the end.

We also have the similar loss landscape visualization (Figure 9) as a side evidence of how different $\alpha$ improves the landscape.

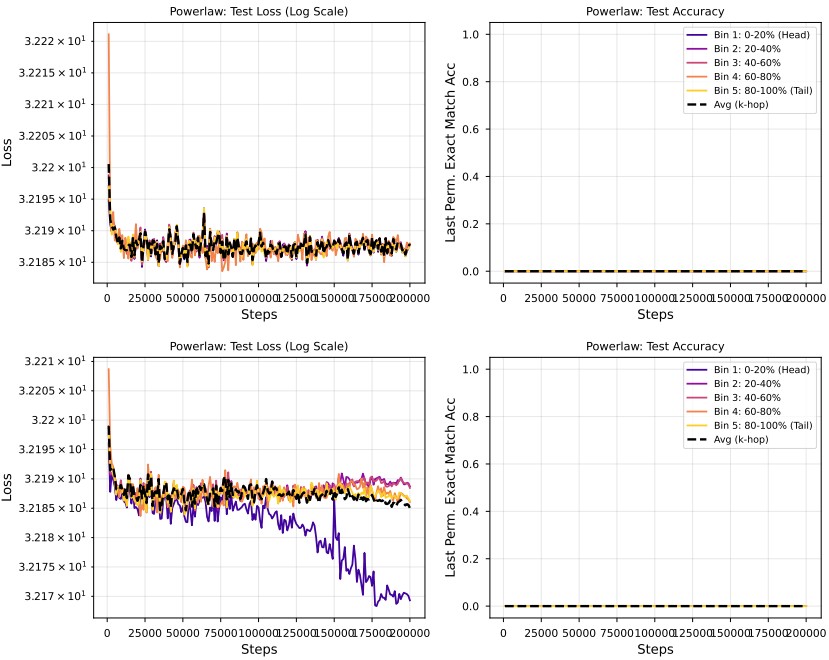

*Figure 7.* The loss curves and accuracy curves for $\alpha = 0.5$ (Top) and $\alpha = 0.75$ (Bottom). When the exponent is not large enough, the optimization is still not benign enough for successful learning of composition.

## C.2. The order of the operation

Because the skills are not all equally difficult, the ordering of skills in the power-law sampling procedure can affect learning dynamics. Some skills are comparatively easy or fundamental, such as $\{1, -1\}$ in arithmetic and the identity permutation in $S_5$. We show in Figure 10 that *the order does matter*: default lexicographical order of the numbers or permutations learn much faster with the same exponent $\alpha$. For example, now we select $\alpha = 1.5$. As shown in Figure 10, the lexicographical order case learns slightly faster than the random rank. We also report that while a power law distribution with $\alpha = 1.0$ works for lexicographical order enables transformers to learn composition, models cannot learn composition with only $\alpha = 1$.

However, *power law still significantly helps optimization* under a **random** order of permutations with an appropriate $\alpha$. Based on this initial experiments, the asymmetric power law still accounts for the improvement that makes the state tracking composition task learnable. We also tried the reversed lexicographical experiment in Figure 10 and the results are similar. We conjecture that the advantage can be strengthened by a designed/structured order of skills.

## C.3. Compatibility with curriculum

We find that the power law distribution aligns naturally with curriculum learning and may offer a stronger training strategy. As described in Section C.3, we adopt an explicit curriculum over hop counts $k = 1, 2, 3, 4$, following Wang et al. (2025), so that supervision progresses from easier to harder examples as task complexity increases. Beyond the uniform baseline, we additionally test a power-law distribution to examine this hypothesis. The experimental results indicate that, even with curriculum learning, uniform training continues to show noticeable plateaus in the loss curve. By comparison, the power law distribution further facilitates optimization, significantly mitigating these plateaus and accelerating training, likely by improving the loss landscape throughout the optimization path.

## C.4. The granularity of the asymmetry

Our results show that power-law distribution is a sufficient condition for successful training on compositional reasoning tasks. Although the analysis does not rule out other asymmetric distributions that enable LLMs to acquire composition capabilities, we conjecture that *better 'granularity of asymmetry' may lead to better training loss landscape, which further accelerates training.* Power law is an example of fine-grained asymmetry.

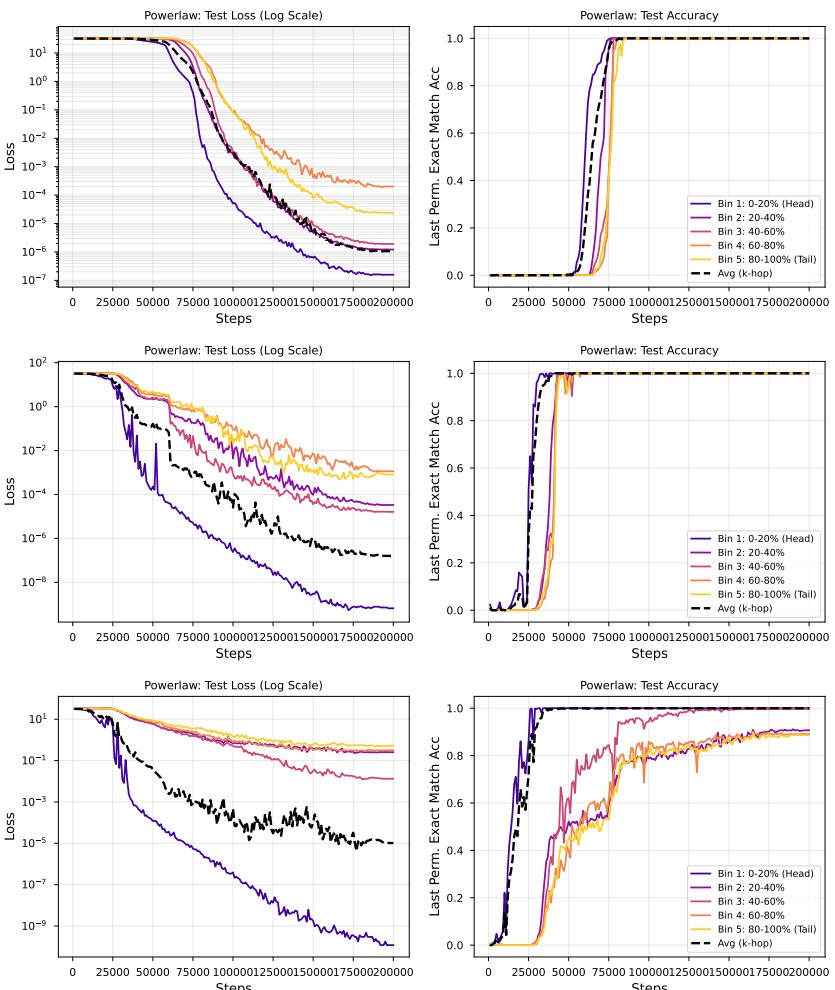

*Figure 8.* The loss curves and accuracy curves for $\alpha = 1.0$ (Top), $\alpha = 1.25$ (Mid) and $\alpha = 1.5$ (Bottom).

Here we tried several different, more coarse-grained power-law distributions: we divide the $|S_5| = 120$ permutations into $m = \{5, 10, 20, 40, 60\}$ bins in lexicographical order. Then we assign the sum probability for different bins with the power law over $i \in \{1, 2, 3, ..., m\}$ with the sum probability $P_{i,sum} \propto \frac{1}{i^\alpha}$. Within each bin, we keep the individual skill probability in each bin uniform, so the individual skill split $P_{i,sum}$ evenly within the bin.

The intuition is that the larger $m$ is, the distribution is more fine-grained and closer to original power law. Here we pick $\alpha = 1.5$ for better learning speed. The experiments are shown in Figure 12, the more fine-grained the distribution is, the faster the model learns.

## D. Multi-hop QA

We followed (Yao et al., 2025a) to construct the natural language multi-hop QA task. Comparing with arithmetic tasks, the QA task is more knowledge-heavy and with a slightly simpler structure. (Yao et al., 2025a) found that the model needs exponentially many $k$-hop data for transformers to learn.

**Dataset**  The dataset contains $|E|$ entities—each with a unique name—and $N$ relation types. We created $|E|$ distinct single-token person names (e.g., Jennifer) and $|\mathcal{R}| = 20$ single-token relation names (e.g., instructor) to serve as namespaces for entities and relations. We reused the name list in (Yao et al., 2025a). The complete list of relation names and a partial list of entity names appear in Tables 5 and 6 in (Yao et al., 2025a). The multi-hop questions are generated through a graph with $|E|$ individuals. Each entity is connected to $|\mathcal{R}|$ randomly chosen person in the graph. We considered the number of

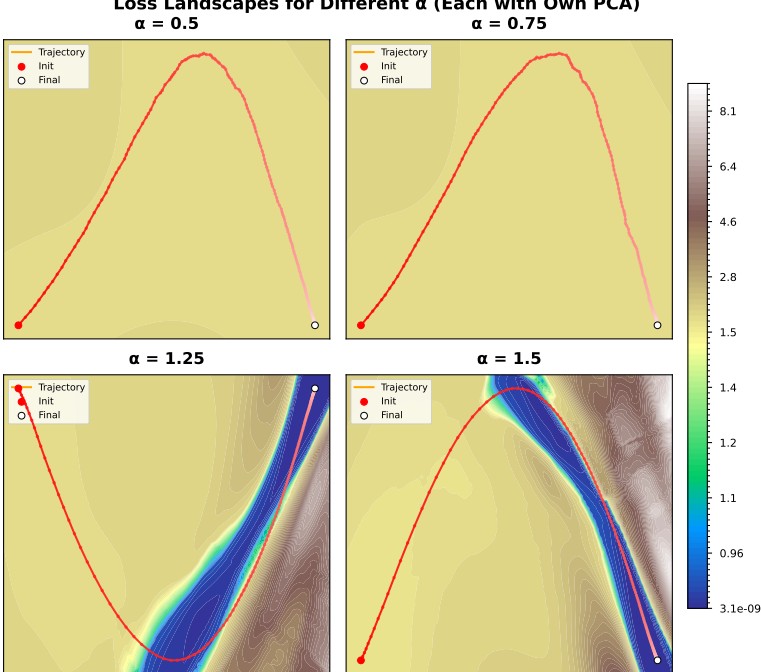

*Figure 9.* Landscape visualization for different $\alpha$s. Larger $\alpha$ has better initial landscapes, while the landscape for small $\alpha$s are still flat.

individuals $|E| \in \{20, 50\}$.

For training, we use online sampled 3-hop and 4-hop training set and test on the leave-out 4096 test questions. For each test instance, we greedy decode the single token answer given the question prompt (e.g. 'Who is the instructor of the teacher of Bob? $\backslash n$ Answer:'). We evaluate the exact match accuracy. The prompt format is as follows:

> **Example: Multi-hop QA**
>
> **Facts:** The teacher of Bob is Carol. The instructor of Carol is Alice.
> **Prompt:** "Who is the instructor of the teacher of Bob? $\backslash n$ Answer:" **Label**. Alice.

**Training details**   We use GPT2-tokenizer with the special tokens like names and relations added to the tokenizer. We use AdamW optimizer with $(\beta_1, \beta_2) = (0.9, 0.98), \epsilon = 10^{-6}$ and gradient clipping 1.0. We run 1000 steps of linear warmup followed by a cosine learning rate schedule to minimal learning rate 0.1 of the peak learning rate. We use bf16 training with packing with context length 1024 tokens. QA pairs from distinct samples are masked from each other during training. We all run the experiments with 3 different random seed and calculate mean and variance.

We use a base model architecture with 384 hidden dimensions, 6 attention heads, and 6 layers. We set the learning rate to 0.0002 with 1000 warmup steps and train for a total of 80,000 steps using batch size 1024, with cosine learning rate decay to $0.1\times$ initial learning rate. We run all experiments across 3 random seeds and report the average performance.

### D.1. Additional experiments

Similar to the $k = 3, |E| = 50$ case in the main paper, the superiority of the power law holds generally across different task settings and random seeds. We set $k \in \{3, 4\}, |E| \in \{20, 50\}$. We re-plot the figure in the main text for completeness in Figure 13. We also did some coarse-grained power law that separates the permutations into 4 groups decided according to the order (in the probability of the power law assigned, see Figure 14).

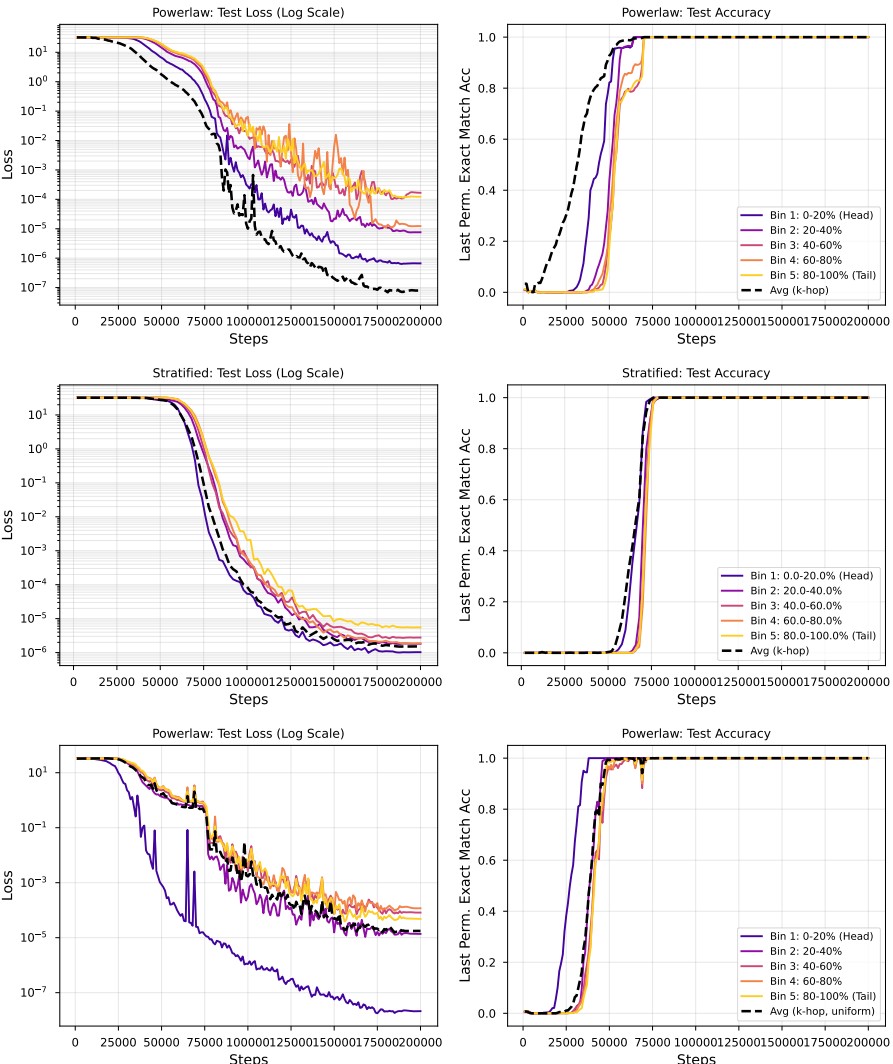

*Figure 10.* The order ablation experiments on $S_5$. The **top** two figures are using lexicographical order and the **middle** two are using random order, which is used in the power-law sampling. The learning process with lexicographical order learns slightly quicker than the random order. Here $\alpha = 1.5$. The **bottom** one is using reverse lexicographical order. The training accuracy of each bin still shows similar performance.

# E. Grade school math problems

**Training details**   We use a standard GPT-2 tokenizer extended with necessary special tokens. We train a decoder-only Transformer model equivalent to GPT-2 Small, featuring 12 layers, 12 attention heads, and an embedding dimension of 768, totaling approximately 124M parameters. We use the AdamW optimizer with $(\beta_1, \beta_2) = (0.9, 0.95)$ and a weight decay of 0.1. We employ a cosine learning rate schedule with a peak learning rate of $5 \times 10^{-4}$ and a minimum learning rate of $5 \times 10^{-5}$ ($0.1\times$ peak), following a linear warmup of 100 steps.

Training is performed in bfloat16 precision with a context length of 1024 tokens and a global batch size of approximately 0.5M tokens ($8 \times 64 \times 1024$). The model is trained for a total of 5 billion tokens, with checkpoints saved every 100 million tokens. We run all experiments across 3 random seeds and report the average performance with standard deviation. For data synthesize pipeline, we directly reuses the structure dependency graph generator in Zhou et al. (2025) and switch to a more natural language template as follows:

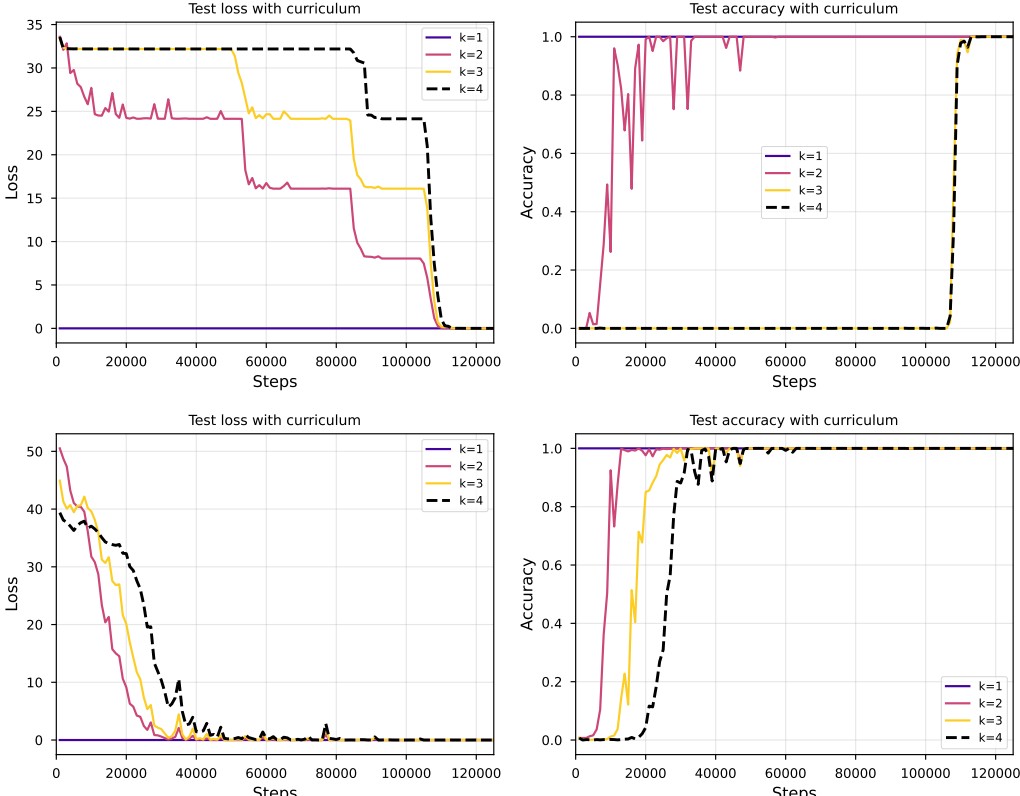

*Figure 11.* (**Top**) Explicit curriculum with mixture of $k = 1, 2, 3, 4$ hops under uniform distribution. The model eventually learns the group elements composition, but the loss plateaus occasionally, suggesting that curriculum alone does not fully remove the optimization difficulty. (**Bottom**) Power law + explicit curriculum further accelerates training and substantially reduces the plateaus. This indicates that power law sampling is complementary to explicit curricula rather than merely duplicating their effect.

---

**Example: Synthetic GSM**

**Problem:** "We are in Mare Serenitatis. There are 18 Eyelash Viper. There are 2 Minke Whale. There are 197 Pelican. There are 2 Forest Mammoth. There are 18 Boomslang. There are 26 Gull. There are 185 Boxfish. There are 160 Dyeing Dart Frog. The total number of Dyeing Dart Frog and Chinstrap Penguin is the quotient of Dyeing Dart Frog and Chinstrap Penguin. ... We know the total number of Dyeing Dart Frog, Chinstrap Penguin, Boxfish, Gull, Boomslang, Forest Mammoth, Pelican, Minke Whale and Eyelash Viper is 18. What is the Chinstrap Penguin?"

**Solution:** We know the total number of Dyeing Dart Frog, Chinstrap Penguin, Boxfish, Gull, Boomslang, Forest Mammoth, Pelican, Minke Whale and Eyelash Viper is 18. We know the Eyelash Viper is 18. Total number of Dyeing Dart Frog, Chinstrap Penguin, Boxfish, Gull, Boomslang, Forest Mammoth, Pelican and Minke Whale is 18 - 18 = 0. We know the Minke Whale is 2. Total number of Dyeing Dart Frog, Chinstrap Penguin, Boxfish, Gull, Boomslang, Forest Mammoth and Pelican is 0 + 2 = 2. We know the Pelican is 197. Multiplying 2 by 197 gives 394... We know the Dyeing Dart Frog is 160. Splitting 160 evenly into 20 parts gives 8, which is the Chinstrap Penguin. Answer: #### 8
**Label**. 8.

---

The operation number is limited within $\{2, 3, ..., 8\}$. The test set is sampled from data with uniform distribution. **The skill here is the number $\{1, 2, 3, ..., \max p\}$.** (1) When we allow modular arithmetic, we directly sample the number from 0 to $p$. It will exhibit a perfect power law curve. We consider $p = 211$. (2) When we use basic arithmetic, we sample the number from 0 to $p$, but reject the sampling when the answer exceed 1000 or cannot be divided. That will upsample some small numbers like 1 to 10, but the rest will still follow power law distribution.

### E.1. Additional experiments

The additional experiments are (1) using a multi-hop template where allows non-rigorous combination of adjacent steps. (2) use basic arithmetic without modulo $p$. In all settings, power-law significantly performs better/train faster than uniform. See Figure 15.

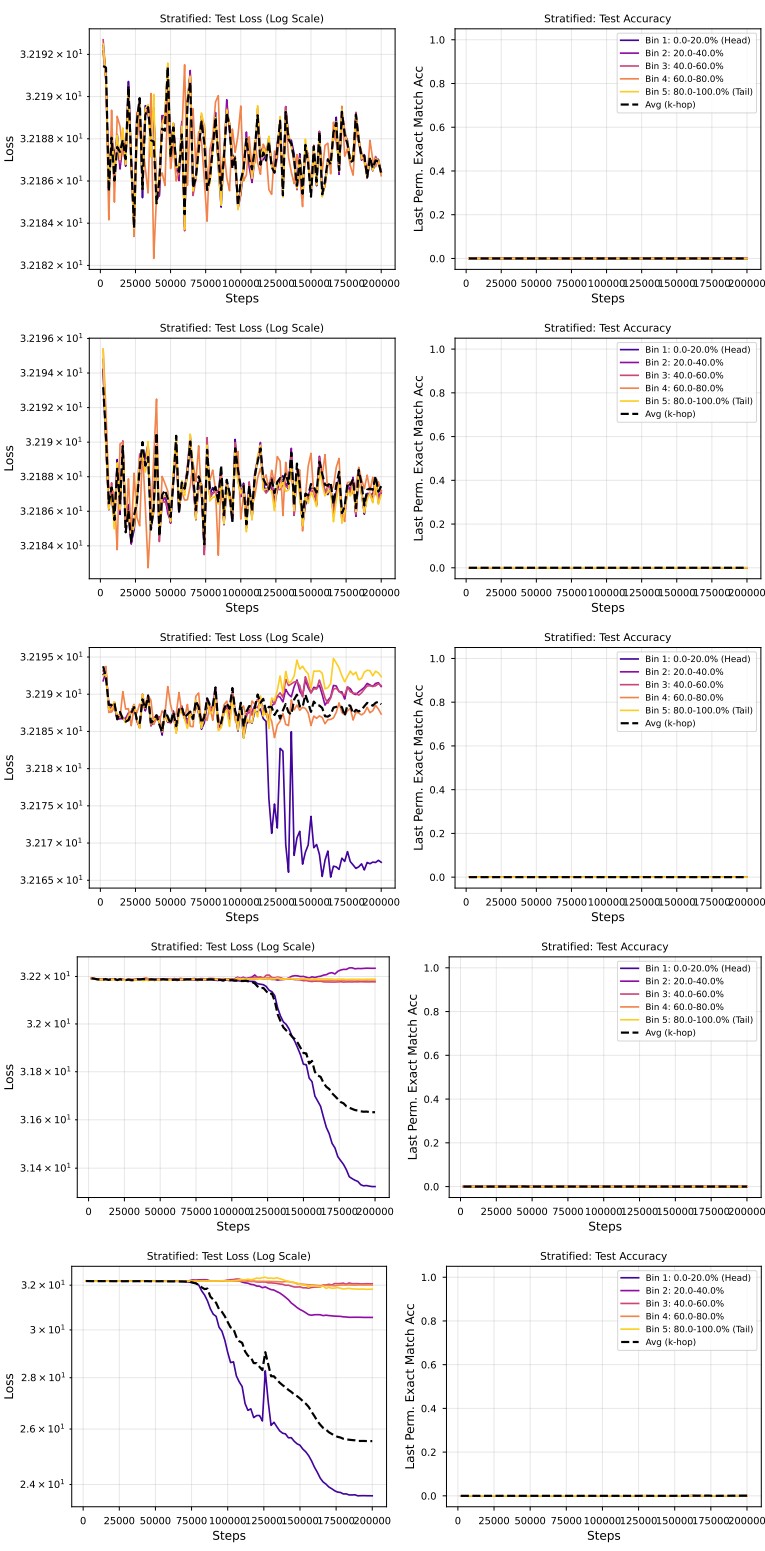

*Figure 12.* The granularity ablation experiments on $S_5$. From top to bottom are # of bins = $\{5,10,20,40,60\}$. When # of bin = 120, it falls back to the original power law. Here $\alpha = 1.5$. As shown in the plot, coarse-grained power law learns much slower compared to fine-grained power law. We conjecture that the fine-grained asymmetry is the key to improve the landscape when the task is intrinsically symmetric and many saddle points exist.

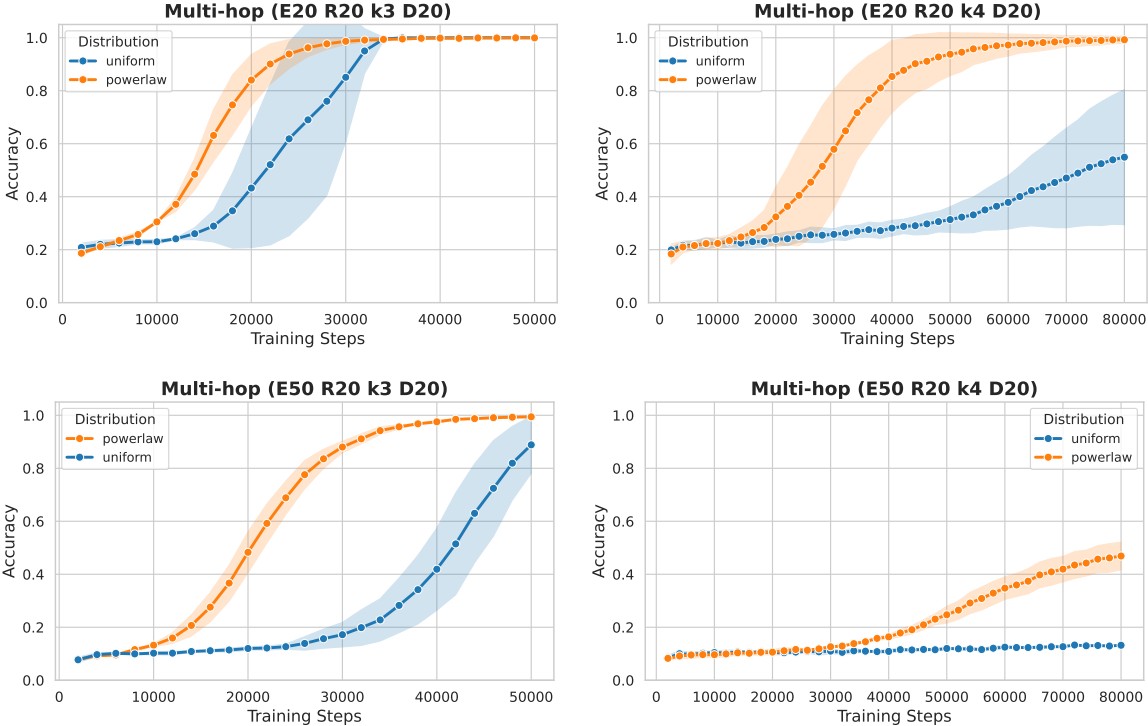

*Figure 13.* Accuracy plots for the multi-hop QA task. Across different data settings, power-law distribution generally accelerate the learning of such multi-hop natural language reasoning tasks. The difficulty indeed increases when the hop number $k$ and individual number $|E|$ grows, but power law always help in terms of training.

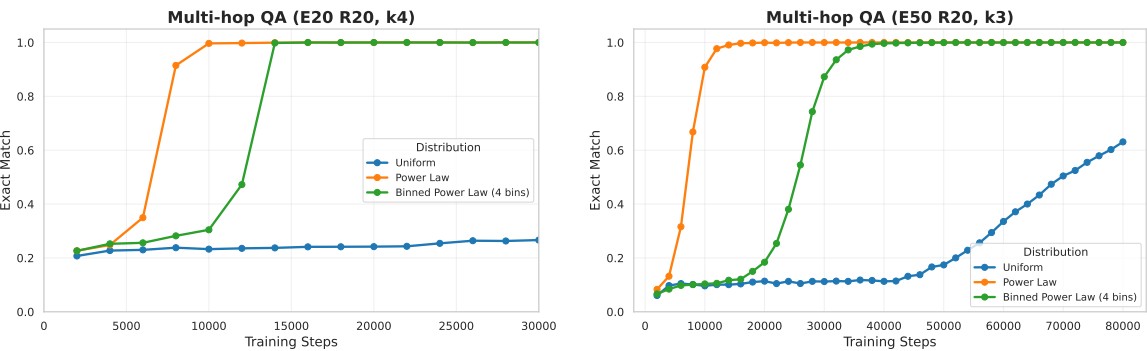

*Figure 14.* Test accuracy in the multi-hop QA tasks. Training with fine-grained power law outperforms coarser-grained (binned) power law and uniform distribution.

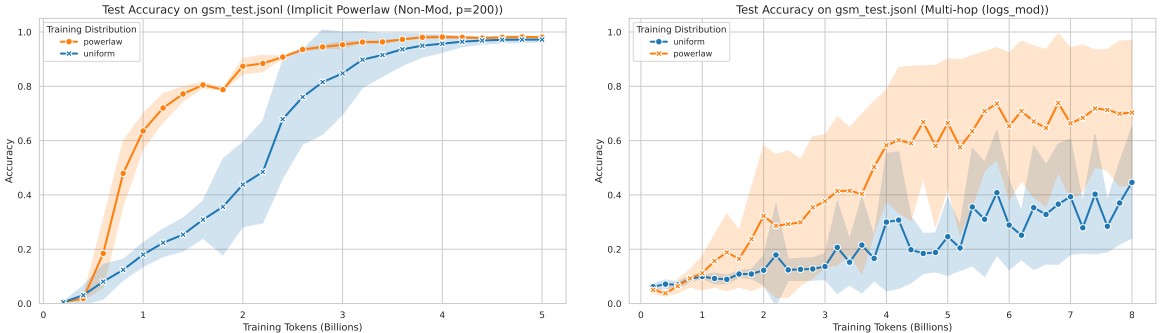

*Figure 15.* Accuracy plots for GSM tasks. **Left**: non-modular arithmetic with maximum leaf value $p = 200$. **Right**: modular arithmetic with $p = 211$, but with multi-hop template randomly combine two steps. Power-law distributions generally helps the model to learn to solve Grade school math synthetic problems much faster than uniform distribution.

