# OpenReview forum: "The Power of Power Law: Asymmetry Enables Compositional Reasoning"
_ICML.cc/2026/Conference — ICML 2026 spotlight_

### Official Review · Reviewer_MB3Z · 2026-03-04

**Soundness:** 4
**Presentation:** 3
**Significance:** 4
**Originality:** 3
**Overall Recommendation:** 5
**Confidence:** 5

**Summary:**

This paper studies the impact of the data distribution on performance on compositional tasks inspired by reasoning. In particular, given a task whose solution requires the combination of a given set of "skills", the authors find that having a Zipfian distribution of skills in the training data can improve performance, contrary to the intuition suggesting that a uniform distribution speeds up learning. Starting from a simple illustrative experiment, the authors define a toy compositional task where the advantage of the Zipfian distribution can be characterised rigorously, then provide a mechanistic interpretation of their result based on the properties of the loss landscape, and finally show that the effect uncovered takes place also in natural language reasoning tasks.

**Compliance With Llm Reviewing Policy:**

Affirmed.

**Final Justification:**

See the Significance & Originality section for the motivation

**Key Questions For Authors:**

1. I am curious as to why the authors picked those specific values of $\alpha$ for Figure 1 (see also the soundness section above).

2. Is there a typo at the beginning of the "key proof idea" paragraph on page 4? "which is the expectation of the training loss in expectation".

3. The authors insist on $\alpha>1$. I understand where that comes from, but I think that $0<\alpha<1$ would cause the probability of head skills to vanish as some inverse power of $d$ between $0$ (corresponding to $\alpha=1$) and $1$ (corresponding to $\alpha=0$). Following the remark on page 5, bottom of the left column, could this create some interesting additional tradeoff?

**Limitations:**

The authors adequately discussed the limitations. No particular potential negative societal impacts are foreseen.

**Strengths And Weaknesses:**

## Soundness

**Strengths.** All the claims are clearly formulated and supported by both theoretical analysis on a toy task and experiments. I don't see any flaw in the experimental designs. I believe that the scope of the experiments is sufficiently broad for acceptance.

**Weaknesses (minor)** However, I also believe that a broader exploration of the roles of the Zipf exponent $\alpha$ and the number of compositional steps $k$ could significantly strengthen the manuscript. In particular, following the comparison with Yao et al. (2025), it would be interesting to empirically characterise whether Zipfian data change the scaling with $k$. The theory says that, with the right $\alpha$, the sample complexity should not depend on $k$, but this is not the case for multi-hop reasoning tasks. Do the authors have a sense of why?

## Presentation

**Strengths.** The submission is very well structured and easy to read. The initial simple experiments help build up intuition, and the mathematical results are (mostly) presented in a clear logical order, with sufficient intuitive explanations that help grasp their implications.

**Weaknesses.** I found the first explanation of the $k$-multiplicative composition task, in section 3.1 of page 3, confusing. The initial phrasing of the "setting" paragraph suggests that there are $k$ skills, while I understand that there are $d$ available skills in total, while each datum requires using $k$ to answer correctly.  The next paragraph begins with "the goal of the task is to uncover the hidden scalar", but to me it looks more like a parity task in skill space. Perhaps presenting the task like a parity in skill space would be more effective. A second minor point: I don't like the phrase "constantly large". The only property required of the probability of `head' skills is that they are not vanishing for large $d$, or 'constant'. Another option could be 'finite', as opposed to infinitesimal, for large $d$.

## Significance & Originality

The contribution of this paper is significant in that it explains, with both theory and experiments, that training under power-law distributions can be beneficial for sample complexity. The main idea behind the result is quite simple and general, hence it could be applied in different theoretical settings, and also inform the choice of training distribution/curricula in practically relevant settings. While the mathematical analysis follows standard techniques, it provides novel insight. If anything, the simplicity of the mathematical ideas makes the result easier to understand.

## Summary

The presentation weaknesses are the only reason I am scoring the submission 4 and not 5. I will raise the score if such weaknesses are addressed.

---

> ### Author Rebuttal · Authors · 2026-03-31
>
> We sincerely thank the reviewer for their insightful comments and questions, especially the comments on our presentation. Due to the ICML policy, we cannot modify the current submission, but **we are committed to improving our writing in the next revision, such as adding more explanations for our setup and correcting typos throughout the paper.** We address each point below and welcome further discussion.
>
> W1: Thanks for the great question!  Here we first answer the question on $k$, and defer the discussion on $\alpha$ later when answering Q1 and Q3 together. Empirically, the difficulty increases with $k$ as noted in Yao et al (2025), and our experiments in Figure 10 also showed that the training becomes slower for both uniform and power law distributions.
>
> For the theory, the sample complexity actually depends on $k$ even though it does not appear on the exponent of $d$. Since $k$ is $\Theta(1)$ and $d\gg 1$, we write the sample complexity in the form of a polynomial of $d$. For simplicity and clarity, we hide the constant terms with $k$ in the $O(\cdot)$ notation. If $k$ is also taken into account, the sample complexity should be $O(c^kd^{2\alpha})$ v.s. $d^{\Omega(k)}$ where $c=\Theta(1)\ll d$ is a constant that only depends on $\alpha$. Intuitively, this is a necessary price to pay to escape a $k$-th order saddle. That means the sample complexity for the power law also increases with $k$, but the rate is much slower. Similar sample complexity separations can be found in the learning multi-index model literature (e.g., [1] Damian et al. https://arxiv.org/abs/2206.15144). We will also add this remark in the appendix.
>
> W2: Thanks to the reviewers for pointing the presentation issues out! The task is exactly similar to parity in the skill space, as the reviewer pointed out. We also discussed the connection with parity at the end of the section. The initial phrasing of the setting is actually suggesting “composing the sequence of $k$ skills, where each skill is **sampled from $d$ available skills**”. We totally agree that we should use ‘constant’ to describe the property of the probability of the head skills. We will improve the presentation in the future version as requested, and happy to take further suggestions.
>
> Q1: Thanks for pointing this out. By default, all the exponents in our experiments are chosen as $1.0$ to match Zipf's law. However, the task $S5$ with random order of permutations (which ablates the effect of different difficulties in the group elements to stress the effectiveness of the power law) is quite hard to train with $\alpha = 1.0$. So we increased the exponent to $\alpha =1.5$, which enables the training. This is also discussed in Appendix D.3. Other experiments in S5 with lexicographical order (default) all use $\alpha=1.0$.
>
> Q2: Thanks for pointing out the typo! We will fix that in the next version.
>
> Q3: A very good and precise question again! In the theoretical part, we insist $\alpha > 1.0$ to ensure the head probability to make sure the probability of head skills is constant without vanishing, exactly as the reviewer pointed out. If $\alpha$ is in $(0,1]$, there could be some potential for further improving this specific theoretical upper bound. Specifically, it is a tradeoff between the first stage and the last long-tail stage (stage 3). If $\alpha < 1$, there won’t be an O(1)-time initial escape because the initial signal vanishes, though this will benefit in the last stage. The initial phase will require $d^{\Omega(poly(k))}$, which is asymptotically worse. If $\alpha =1.0$, the initial alignment becomes $1/\log d$ and will have some asymptotic improvement (e.g. $d^2\log^k d$) if we ignore all polylog(d) factors.
>
> However, empirically the training speed always increases with the exponent $\alpha$ (maximum 1.5) when the batch size is reasonable (e.g., less than 2048). We conjecture that in more realistic settings, stage 1 is always the bottleneck. So we decide not to dive too deep on the best possible theoretical bound and insist on $\alpha > 1$.

---

> > ### Author Rebuttal · Reviewer_MB3Z · 2026-04-03
> >
> > Thanks! All my concerns have been addressed, and I believe that the proposed modifications will improve the presentation. I will raise my mark to 5. This paper clearly merits acceptance in my view.

---

### Official Review · Reviewer_ZEfF · 2026-03-06

**Soundness:** 4
**Presentation:** 4
**Significance:** 4
**Originality:** 3
**Overall Recommendation:** 5
**Confidence:** 4

**Summary:**

In this paper, the authors show that training under power-law distributions outperforms training under uniform distributions, contrary to the common belief that reweighting data toward a uniform distribution facilitates learning long-tail skills that appear less frequent. The authors first show two motivating experiments on compositional reasoning tasks, where training under power-law distributions outperforms training under uniform distributions by a relatively large margin. The authors then present theoretical results and mechanistic analysis behind this phenomena, noting that power law data distribution makes it easier to escape the flat region in the loss landscape. The authors also show ablation studies on the exponent in power law data distribution. Lastly, the authors validate the advantage of power law data distribution on two synthetic natural language tasks, Multi-hop QA and Grade school math problems.

**Compliance With Llm Reviewing Policy:**

Affirmed.

**Final Justification:**

This is a strong, well-executed and well-written paper that provides important insights about the relationship between training data distribution and learning less frequent skills.

**Key Questions For Authors:**

- Experimental results are very strong across a wide range of benchmarks presented in the paper. I would like to understand more about the potential limitations of power-law data distribution; perhaps it is possible that power-law data distribution boosts performance on compositional reasoning tasks, but harms performance on knowledge-based tasks or creative writing tasks? This point was briefly mentioned in the conclusion section, but if you could show the performance of models you trained on non-compositional reasoning benchmarks, I think it could make the results even stronger.

**Limitations:**

yes

**Strengths And Weaknesses:**

This paper is pretty strong overall, providing novel and important insights about the relationship between training data distribution and learning less frequent skills. The experiments are also very comprehensive (and with strong results), although the evaluation is mostly limited to language models of size similar to GPT2, and most tasks are synthetically designed. Mechanistic analysis of training dynamics under power-law data distribution also provides interesting insight into why power law distribution helps learning long-tail skills. Presentation is also quite strong, with many high quality figures; In particular, Figure 1 effectively captures the core message of this paper.

---

> ### Author Rebuttal · Authors · 2026-03-31
>
> We sincerely thank the reviewer for their insightful and encouraging acknowledgment of our contribution and presentation. We are happy to take your question and intend to make the results even stronger.
>
> **Q1.** Thank you for the insightful suggestion! We believe that a power-law distribution will indeed hurt performance on knowledge-based tasks, as a heavy-tailed distribution leads to undertraining on the tail and thus degrades memorization. We also conducted an additional experiment on single-hop (atomic knowledge) QA and profiles on the multi-hop QA task setting (in the anonymous link, https://anonymous.4open.science/r/ICML-PowerLaw-Rebuttal-2503/power_law_composition.pdf (Figure 4), which shows that **the power-law distribution begins to fall short of the uniform distribution** (knowledge-based, non-compositional task). This is indeed consistent with the intuition, and echoes many previous papers showing that the heavy-tail effect harms learning knowledge. We would be happy to add further experiments if the reviewer could kindly suggest relevant prior work on creative writing tasks.

---

> > ### Author Rebuttal · Reviewer_ZEfF · 2026-04-01
> >
> > Thank you for adding these additional experiments! I will maintain my initial positive rating.

---

### Official Review · Reviewer_G1Nw · 2026-03-10

**Soundness:** 3
**Presentation:** 3
**Significance:** 3
**Originality:** 2
**Overall Recommendation:** 5
**Confidence:** 3

**Summary:**

This paper studies the impact of the data distribution on the learnability of compositional reasoning tasks. In particular, they compare a power-law distribution (a characteristic of natural data) over skills to a uniform distribution, and find that models learn faster and better with the power-law structure.

The benefits of the power-law are demonstrated both theoretically and empirically. On a simple k-multiplicative composition task, the authors bound the number of data samples necessary to learn the task to a given precision, and prove the superiority of power-law when the composition number (i.e. how many elements to compose) is large compared to the power-law exponent. Based on some elements of the theory, they argue that the asymmetry in the power-law structure creates sharper optimization landscapes, creating an implicit curriculum over skills that accelerates learning in the early to middle stages, but slows down learning of the remaining (rarest) skills towards the end of the optimization process. Experiments on natural language reasoning tasks show that the benefit of the power-law distribution carries over to this setting too.

**Compliance With Llm Reviewing Policy:**

Affirmed.

**Final Justification:**

The final clarifications and  experiments now fully address my concerns, and they strengthen the overall message of the paper. I therefore updated my score to Accept (5).

**Key Questions For Authors:**

Questions

The authors state
> though larger $\alpha$ leads to faster training in the head, the tail skills’ learning will be slowed down due to smaller sampling probability. The result echoes our theory and indicates that there is a trade-off on the exponent $\alpha$.

Could the authors elaborate on which parts of this finding is explained by the theoretical results?

**Limitations:**

yes

**Strengths And Weaknesses:**

Strengths
- **Relevance**: The studied problem (the data distribution’s influence on compositional reasoning) is interesting and relevant to the ICLR community.
- **Soundness of the theory**: The theoretical results appear reasonable (I did not check the proofs in detail), and the terminology is explained well.
- **Presentation**: I found the paper well-written and clear.
I really enjoyed the loss landscape visualizations in section 4.2, and the interpretation of ‘symmetry breaking’. Despite not being a fully formalized theorem, I think this section does a good job at explaining overall trends in an intuitive way, but supported by experiments and some mathematical arguments.
- **Originality**: The paper builds on past work in the tasks they study, but the framing (power-law vs uniform) appears new. The proofs appear novel.


Weaknesses
- **Significance**: I think that using a uniform skill distribution as a baseline is not well-motivated and even slightly counterintuitive for compositional tasks. Intuitively, fully uniform sampling is most plausible when the individual objects are (on average) independent and of similar difficulty. This is not the case in the compositional reasoning tasks considered: e.g. the skills x3 and x6 are related. I agree with the authors that there is a controversy in the literature on whether data should be balanced or not. Section 4.2 (learning stages) points out a tradeoff stating that while power-law is better overall, very rare skills are much slower to learn. My intuition is that natural data is structured in a way that the rarest skills / elements of knowledge are often either too complex (and hence ‘unlearnable’ even with more occurrences), or less related to other skills/pieces of knowledge, hence missing them completely can still be better than learning them to a limited amount via the uniform weighting. Overall, I think that rather than demonstrating a surprising phenomenon (that power-law outperforms uniform), the paper confirms an a-priori intuitive hypothesis, which limits its significance.
- **Insufficient/incpmplete Experiments**
    - There are few baselines apart from the uniform considered, and these only appear as ablations in the toy setups (i.e. the coarse-grained power-law distributions in ‘the granularity of asymmetry’). These experiments are called ‘initial’, but it is not clear what makes these results ‘initial’. I think a more structured comparison of baselines extending to natural language tasks would strengthen the paper.
    - The relationship to curriculum should be discussed more. The order ablation experiments (as stated in the paper) are not yet conclusive. While results showing the benefit of a structured order are claimed, the referenced Figure 9 only contains results for the $S_5$ task and not the arithmetic task. Also, the caption is confusing: it mentions the left and right figure when there are 4 subplots, and the plot titles also do not specify which is the lexicographical order task. In section D.3., there appear to be claims of findings that are not presented quantitatively and called ‘preliminary’. It would be interesting to study the effect of explicit vs implicit curriculums (via the power law), for instance by controlling the correlations between skills and $\alpha$ jointly.
    - Some additional experiments (App F.1.) only have their results stated qualitatively in words, but not demonstrated with a figure/table. I recommend the authors to add these demonstrations to establish the credibility of the results.

Recommendations

I think the paper makes a great start, but certain aspects are unfinished. I recommend the authors to:
- Instead of ‘presenting a surprising result’, I would position the paper as an investigation of **how** the power-law structure aids the learning of compositional reasoning tasks.
- Re-evaluate the extent of the validity of the uniform skill distribution as a baseline, in particular in relation to the ‘datapoint difficulty’ literature (e.g. Sorscher et al., 2022)
- Extend the evaluation of less simplified baselines (such as the coarser power laws)
- Isolate the effect of curriculum / order in a controlled setup (controlling both $\alpha$ and correlations between skills).

---

> ### Author Rebuttal · Authors · 2026-03-31
>
> We sincerely thank the reviewer for their acknowledgement of the relevance of the paper and their insightful comments and questions. We address each point below and welcome further discussion.
>
> **W1(Significance).** Thank you for the constructive comments. First, we would like to clarify our power-law strategy as we find that the reviewer’s intuition does not seem to apply here. In most of our experiments (except Section 4), we use a random ordering of all skills, which should remove the correlation between skill difficulty and the power-law probability.
>
> We also argue that our setting indeed falls into the case “when the individual objects are (on average) independent and of similar difficulty”, and thus uniform skill distribution is a well-motivated baseline. We would like to clarify that, in **both our theoretical and empirical experiments, the compositional tasks we consider involve nearly independent sampling and components of similar difficulty.** For example, all relations in our multi-hop QA tasks are completely symmetric and with the same difficulty, and different relations in each sample are all independently sampled. The numbers and the group elements in S5​ are also of similar "difficulty," except for a few basic elements such as {0,1,−1} in arithmetic and the identity permutation in S5​. We believe the tasks we consider fall under your intuition, and therefore, our findings can be seen as counterintuitive.
>
> Last but not least, to the best of our knowledge, most recent papers that aim to understand natural language via synthetic tasks (including many multi-hop reasoning tasks) use a uniform distribution over "skills" for training [1,2,3,4]. We would be happy to discuss the paper's positioning further if you have additional questions about the compositional reasoning settings.
>
> **W2 (Insufficient/incomplete experiments)**
> - **W2.1** Thank you for the constructive suggestions. In the anonymous link (https://anonymous.4open.science/r/ICML-PowerLaw-Rebuttal-2503/power_law_composition.pdf), we added experiments with coarse-grained power-law distributions in multi-hop QA tasks and show similar results in Figure 1. This further suggests that finer-grained power laws may also benefit training on natural language tasks. In this setting, we also plot the loss landscape and the test loss for different ranks of relations in the QA tasks (Figure 2 in the link), which again confirms behaviors similar to those in the state tracking task. We hope these additional results address your concern.
> - **W2.2** Thank you for pointing out the presentation issues in the appendix. The caption of Figure 9 should say "the two figures on the left" and "the two figures on the right." Regarding your last question, we used a random ordering of permutations in Appendix D.3, which should decouple the correlation between skill difficulty and the power-law probability. We further conducted experiments with 5 different random seeds for the ordering of all permutations with $\alpha = 1.5$, and they all show similar results. We also added more arithmetic results to the link. We believe this provides additional evidence for the effectiveness of the power-law distribution, apart from the effect of explicit curriculum. We are not sure whether this fully addresses your concern, and would be glad to take more specific suggestions on controlling explicit vs implicit curricula.
> - **W2.3** Thank you for pointing out this mistake. The figure referenced in F.1 should be Figure 11 in the paper, and we apologize for not placing it directly below the subsection title. We are sorry for the confusion. We will reformat the figure and add a reference link in a future version.
>
> Regarding the recommendations: Thank you for your constructive and helpful suggestions and we have settled them in previous rebuttals. We would be happy to add more discussion in the related work section regarding the data point difficulty (e.g. Sorscher et al., 2022) in the next version. Although the literature studies a different setting from compositional reasoning, it is very interesting and indeed related to our work. We would be glad to answer any further questions if anything remains unclear.
>
> **Q1.** In our theory, a larger $\alpha$ leads to asymptotically longer convergence time in Theorem 2, where the convergence time scales as $t \sim d^\alpha$. This is exactly because tail skills are learned more slowly, as described in our Stage 3 argument in Section 4.2. The faster training on head skills can also be explained by Stage 2 in Section 4.2, although we did not state this explicitly. Specifically, a larger $\alpha$ assigns a larger probability $p_i$​ to head skill i (proportional to $1/i^\alpha$), which leads to a stronger gradient signal according to the gradient formula.
>
> [1] Liu et al. https://arxiv.org/abs/2210.10749 [2] Ye et al. https://arxiv.org/abs/2407.20311 [3] Allen-Zhu et al. https://arxiv.org/abs/2512.17351 [4] Li et al. https://arxiv.org/abs/2402.12875

---

> > ### Author Rebuttal · Reviewer_G1Nw · 2026-04-02
> >
> > Thank you for the rebuttal, and the great additional experiments! My concerns, apart from the explicit curriculum are addressed.
> >
> > In my suggestion about testing the role of curriculum, I meant an extension of Appendix D.3, where you tested the implication of using the standard lexicographic order for permutations vs a random ordering. It seems that lexicographic order induces a curriculum of increasing complexity, which may be the reason why this order especially encourages compositional learning. It seems to me that power law itself creates an implicit curriculum that forces higher-probability skills to be learnt first, and learning the simplest skills first might transfer better to other skills. It would be great if the authors could ablate how these explicit and implicit curricula interact with each other. For example,
> > * if permutations are presented in the _reverse_ lexicographic order, does the benefit of the power law decrease?
> > * compared to the uniform baseline, is the power law better at incorporating explicit curricula (e.g. simpler -> more complex)?
> >
> > I think these experiments would strengthen our understanding of the benefits of power law and its interaction with curricula.

---

> > > ### Author Response · Authors · 2026-04-04
> > >
> > > We sincerely thank the reviewer for the thoughtful follow-up and suggestions. We believe the follow-up question involves two distinct notions of curriculum, and we address both below.
> > >
> > > **(1) Clarification on Appendix D.3.:**
> > >
> > > In Appendix D.3, the “order” of permutations does not refer to the temporal order in which training examples are presented. Rather, it only defines how skills are indexed when assigning probabilities under the power-law distribution $p_i \propto i^{-\alpha}$. Training examples are always sampled i.i.d. from the resulting fixed distribution.
> > >
> > > Therefore, under the uniform distribution, the indexing order (lexicographic / random / reverse lexicographic) has no effect at all: every skill appears with the same probability, so the training distribution is unchanged by re-indexing. In this sense, an explicit curriculum is never applied under the uniform baseline.
> > >
> > > To make this concrete in the $S_5​$ task, let the 120 group elements be denoted by $g_1,…,g_{120}​$. Each input is a sequence of independently sampled elements $g^{(1)},...,g^{(4)}$, and the target is their composition $g^{(4)} \circ g^{(3)} \circ g^{(2)} \circ g^{(1)}$. Under uniform sampling, each element appears with probability 1/120 at each position, regardless of how we index the 120 elements. The indexing order matters only under power-law sampling, because it determines which skills receive higher or lower probability mass.
> > >
> > > **(2) Follow-up experiment on the reviewer’s specific concern.**
> > >
> > > We added a reverse lexicographic ordering in Figure 5 of the anonymous link (https://anonymous.4open.science/r/powerlaw_distribution_icml2026-8C1E/rebuttal-reply.pdf). We find that power law sampling still enables compositional reasoning and gives a similar benefit to the random-order case. This suggests that the gain is not tied to a particular lexicographic assignment of high-probability mass, but is robust to how skills are indexed.
> > >
> > > **(3) Interaction with an explicit curriculum.**
> > >
> > > We believe the reviewer’s broader question, “whether power law sampling interacts favorably with an explicit easy-to-hard curriculum,” is important. To study this separately from the indexing issue above, we additionally ran experiments with an explicit curriculum based on the number of hops k=1,2,3,4, following Wang et al. 2025(https://arxiv.org/abs/2505.23683), which creates an easy-to-hard supervision signal over task complexity.
> > >
> > > In that setting (Figures 6 and 7 in the anonymous link https://anonymous.4open.science/r/powerlaw_distribution_icml2026-8C1E/rebuttal-reply.pdf), we observe that uniform + curriculum still exhibits noticeable loss plateaus during training, suggesting that curriculum alone does not fully remove the optimization difficulty. By contrast, power law + curriculum further accelerates training and substantially reduces the plateaus. This indicates that power law training distribution is complementary to explicit curricula.
> > >
> > > Overall, our interpretation is that Appendix D.3 rules out the possibility that the benefit comes from a special lexicographic ordering, while the additional hop-based experiments suggest that power law sampling can further strengthen explicit curricula.
> > >
> > > **Thank you once again for your time and effort. We believe these experimental results can greatly strengthen our paper, and we would be grateful for your reconsideration of the overall evaluation.**

---

### Official Review · Reviewer_1yEM · 2026-03-13

**Soundness:** 3
**Presentation:** 3
**Significance:** 3
**Originality:** 3
**Overall Recommendation:** 5
**Confidence:** 3

**Summary:**

This paper first finds that the power law distribution of composing skills of the compositional task enables faster and better learning compared to that with uniform distribution in the arithmetic composition task and the state tracking task. Using a minimal form of compositional task (k-multiplicative composition), the authors show that the required sample size for learning the task scales much more efficiently when the skills are sampled from a power law distribution, and also walk through how the differential probability of the skills affects the population gradient. Then, they show three stages of compositional learning in the state tracking task (learning head skills, acceleration of the low frequency skills and the convergence). Finally, they show that the power-law distribution shows effectiveness in realistic reasoning tasks.

**Compliance With Llm Reviewing Policy:**

Affirmed.

**Final Justification:**

I maintain my score in support of the paper. We agreed on some of the possible improvements, and the authors updated the manuscript accordingly which I hope was helpful. The authors also clarified my questions during the rebuttal period.

**Key Questions For Authors:**

- I’m curious about 3 stages analysis in the uniform-distribution cases, and in each extent of those stages are specific to the power-law. As the authors mentioned in the ablation, there might be a natural differential initial gradient from the difficulty of the problem (this follows the notion of ‘implicit curriculum’) and I think there might be a similar stage-like phases happening for the uniform-distribution case, but just slower, and one might also see similar trend in generalization (tasks with easier subpart can be generalized first, and then difficult ones later). Or do they show very different learning curves and generalization patterns?
- In a similar vein, was their any more systematic observations on how sampling order following difficulty - distributional asymmetry in your ablation studies on the order of the skills?
-Following your explanation, I’m expecting the larger k (longer compositional chain) will benefit more from the asymmetry, how does scaling look like between (k) and the training efficiency gap between (uniform vs. power law?)
- Finally, your theoretical explanation relies on a small initialization and the differential flatness of initial loss landscape, which might be very different from finetuning the pretrained models. Have you observed any difference when you introduce power-law sampling for finetuning? The limitation briefly mentioned your focus on pretraining, but I am curious if you have any further comments on this.

**Limitations:**

Yes

**Strengths And Weaknesses:**

**Strengths**
- The work is well structured. The paper asks interesting questions and provides answers in multiple dimensions. It starts from empirical observations in a controlled task (multi-step arithmetics, state-tracking task), provides a theoretical explanation with a simple instance of a compositional task and explains the empirical observations with the theory. Finally, they show that the effect still holds in realistic settings, which is satisfactory.
- The paper is well written and easy to follow. There is a clear flow and the paper walks through each step of the study organically. I appreciate that the authors delivered the key intuition of the theoretical explanation and how it connects to empirical observation.
- As the authors discussed in the related works, they connect the idea of the importance of asymmetry in learning to study compositional learning, which is an interesting and important topic.

**Weaknesses**
- It is always a trade-off of doing theory, but the theoretical setting of k-multiplicative task that the authors use is a contrived setup and does not include all nuanced data properties that could be important for learning a compositional task. To nitpick, if I understood correctly, the task is not sensitive to the order of the input skill sequence (am I correct?), while the nature of many compositional tasks includes the order in which the skills are composed. Furthermore, there is no notion of different difficulties among the skills in the current setting.
- The authors claim that the differential gradient signal due to the probability rank of each skill as a major factor for stage-wise learning and compositional capability of the model. However, as the authors touched briefly, the difficulty of each skill can also make the differential gradient signal. The authors perform the ablation studies, but it is unclear yet how the main results using S5 task in Section 4 controlled for this, given that different operations can have different difficulties, and neither the theoretical setup.
- I found the explanations in Section 4, on how does the power-law distribution can help model learning compositional task, but on the other hand, it does not fully explain or study how compositional task fails in the uniform distribution. I relay relevant questions in the
- More of a suggestion, but referencing could be improved when the existing result or terms have used, for example, line 261 PL condition or line 293 implicit curriculum.

---

> ### Author Rebuttal · Authors · 2026-03-31
>
> We sincerely thank the reviewer for their insightful comments and questions. We address each point below and welcome further discussion.
>
> - **W1:** “The task is not sensitive to the order of the input skill sequence.”
> Thank the reviewer for explicitly pointing out the limitation. Yes, the understanding is totally correct for the toy task in our theoretical analysis, which is a simplified version of a typical order-sensitive compositional reasoning task (e.g., S5 state tracking). We acknowledge the limitation, while the contrived setting is necessary so far to make rigorous analysis possible. However, the theoretical insights of the theory transfer to most of the experiments where all nuanced data properties exist. This suggests that our theory captures some essential aspects of the phenomenon.
>
> - **W2**: We controlled the factor of different difficulty levels by randomizing the rank of each permutation (group elements in S5) with several different random seeds, where the power law distribution still consistently helps the model to learn S5 tasks efficiently. Therefore, the model won’t benefit from a higher frequency of certain easy elements.
>
> - **W3(Q1)** It is a very good question. From the theoretical perspective, we first proved an SQ lower bound showing that training with uniform distribution will require at least $d^\Omega(k)$ samples, which is similar to the lower bound on S5 composition in some previous work (Wang et al. 2025 https://arxiv.org/abs/2505.23683). That explains the failure (inefficiency in training) in uniform distribution with a lower bound. Moreover, in the optimization analysis, we can still extend the stage-wise analysis, though stages 2 and 3 will not be clearly separated. It is because all “skills” are quite symmetric, and there won’t be much heavy-tailed effect or learning speed difference. The slowdown of training is mostly because the initial gradient signal in stage 1 is $d^{-\Omega(k)}$ small, which requires $d^{\Omega(k)}$ steps/samples to escape.
>
> In our experiments on our theory setting and S5, uniform distribution never goes to non-trivial accuracy due to the hardness of the tasks themselves. In our natural language settings, chain-of-thought or other intermediate supervisions are added, so uniform distribution can still learn the task but much slower due to a worse loss landscape. Regarding the difficulties of different skills, we believe “tasks with easier subparts can be generalized first, and then difficult ones later” is reasonable, and it would be a very interesting experiment to add for GSM-like tasks (where an obviously easier subpart should be 0 or 1). Thank the reviewer again for the suggestion.
>
> **W4.** Thanks to the reviewer for pointing those out. We will improve the reference for the existing well-known results in theoretical analysis.
>
> **Q2:** The gap will be larger when $k$ is larger, as we predicted in the theory (the gap is $d^{\Omega(k)} $. It can be shown in Figure 10, where we can observe a larger speed gap in $k=4$ compared to $k=3$. At the same time, the theory will also predict that when the number of skills increases, the gap increases. The experiments also show that when the entity number $E$ increases, the gap between uniform and power law widens correspondingly.
>
> **Q3:** Great question! We didn’t try to finetune on S5 or other synthetic tasks. We tried arithmetic in fine-tuning settings, and there is no significant performance gain. Our conjecture is that if the model already learns single-hop (atomic) knowledge (for example, arithmetic is overtrained), there might be little performance gain. But if the knowledge is new, the performance gain should be huge.

---

> > ### Author Rebuttal · Reviewer_1yEM · 2026-04-02
> >
> > I thank the authors for their time and effort for the rebuttals.
> > I keep my initial score in support of their work. Well done!

---

### Decision · Program_Chairs · 2026-04-30

**Decision:**

Accept (spotlight)

**Comment:**

Paper spots an intriguing phenomenon - when you constrain to small Language Models that are trained on specific composition tasks and tested, usually data mixture consists of enough hard samples (that require longer chaining of task solutions) to train to teach a model to solve these long range tasks.

Paper counter intuitively suggests that power law sampling in task complexity (with more difficult tasks being samples less according to a Zipf law) leads to great training progress. Authors first demo this on synthetic arithmetic and state tracking tasks. Authors argue using the PL condition that the loss landscape is favorable with power law distributions in task complexity for a very synthetic model.

Further, authors demonstrate the effectiveness in multi hop question and answering. Reviewers are very favorable about this paper.

This paper certainly merits the attention of people who prepare datasets for SFT or RL for compositional tasks. However, it would have been better if the authors supported this thesis by taking frontier thinking models of smaller size and fine tuned on compositions tasks and showed results on SOTA compositional tasks benchmark that distribution of task complexity in SFT would matter. That would have been a bit more convincing since it would have had reasonable amount of pre-training data baked in the model.

Independent of the recommendation, one curious question I have is: Pre-training dataset is presumably heavy tailed as well in terms of seeing data about different useful compositional tasks where harder problems are usually rare. So perhaps LLMs already exploit this ? However, the idea at this stage and evaluations presented certainly should lead to interesting future directions based on this work.